

# Dense Hopfield networks in the teacher-student setting

Robin Thériault[1⋆] and Daniele Tantari[2]

**1** Scuola Normale Superiore di Pisa, Piazza dei Cavalieri 7, 56126, Pisa (PI), Italy
**2** Department of Mathematics, University of Bologna,
Piazza di Porta San Donato 5, 40126, Bologna (BO), Italy

⋆ robin.theriault@sns.it

## Abstract

Dense Hopfield networks with $p$-body interactions are known for their feature to prototype transition and adversarial robustness. However, theoretical studies have been mostly concerned with their storage capacity. We derive the phase diagram of pattern retrieval in the teacher-student setting of $p$-body networks, finding ferromagnetic phases reminiscent of the prototype and feature learning regimes. On the Nishimori line, we find the critical amount of data necessary for pattern retrieval, and we show that the corresponding ferromagnetic transition coincides with the paramagnetic to spin-glass transition of $p$-body networks with random memories. Outside of the Nishimori line, we find that the student can tolerate extensive noise when it has a larger $p$ than the teacher. We derive a formula for the adversarial robustness of such a student at zero temperature, corroborating the positive correlation between number of parameters and robustness in large neural networks. Our model also clarifies why the prototype phase of $p$-body networks is adversarially robust.

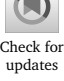

# 1 Introduction

Hopfield networks are artificial neural networks that model associative memory [1]. In the Hopfield model, examples $\sigma \in \{-1, 1\}^N$ of memories $\xi^\mu \in \{-1, 1\}^N$, $\mu = 1, \ldots, M$, are retrieved by sampling the Gibbs distribution of a 2-body Hamiltonian $H[\sigma|\xi]$ at a given temperature $T$ [2]. Hopfield networks can be trained in a biologically plausible way using Hebb's rule [1,3], which leads to $H[\sigma|\xi] = -\frac{1}{N} \sum_{\mu=1}^M \left( \sum_{i=1}^N \xi_i^\mu \sigma_i \right)^2$. However, they can only store up to $M \sim \mathcal{O}(N)$ i.i.d. random memories in the limit of large $N$ [1,4,5]. One way to find this scaling is to study the phase diagram of $H[\sigma|\xi]$ as a function of the temperature $T$ and load $\alpha = \frac{M}{N}$ [5], where the so-called ferromagnetic phase, which extends up to $\alpha \approx 0.14$, corresponds to accurate retrieval.

Since Hopfield's seminal work, several generalizations have been investigated in relation to their critical storage capacity and retrieval capabilities. For example, parallel retrieval has been studied in relation to pattern sparsity [6–10] or hierarchical interactions [11–15], and non-universality has been shown with respect to more general pattern entries and unit priors [16–22]. Efforts to overcome the $\mathcal{O}(N)$ limitation of the capacity led to the development of a novel class of modern Hopfield networks [23–25], which are sometimes called dense due to their faculty to store much more memories than the original Hopfield model [26]. These neural networks surpass $\mathcal{O}(N)$ storage capacity by using higher-order interactions instead of the original 2-body couplings [27–32]. In particular, Gardner [30] calculated the replica-symmetric (RS) phase diagram of the Hamiltonian $H[\sigma|\xi] = -\sum_{i_1 < \ldots < i_p=1}^N J_{i_1 \ldots i_p} \sigma_{i_1} \ldots \sigma_{i_p}$ with $p$-body interactions $J_{i_1 \ldots i_p} = \frac{p!}{N^{p-1}} \sum_{\mu=1}^M \xi_{i_1}^\mu \ldots \xi_{i_p}^\mu$ conditioned on i.i.d. random memories $\xi^\mu \in \{-1, 1\}^N$, finding a $M = \mathcal{O}(N^{p-1})$ storage capacity. These calculations were later extended to include the effects of one-step replica symmetry breaking (1RSB) [33].

Although they draw a rather detailed picture of the retrieval of individual i.i.d. random memories, these results are not the end of the story. First of all, 1RSB calculations allegedly struggle to find the paramagnetic to spin-glass phase transition accurately at large $p$ because of numerical instability issues [33]. Second of all, dense Hopfield networks have been rapidly gaining a renewed attention for reasons other than their storage capacity since a recent paper [26] by Krotov and Hopfield (K & H), where they were used as a trainable machine learning architecture. For instance, they have been related to transformers [23,34] and diffusion models [35,36], and they were found to be significantly more explainable and adversarially robust than feedforward neural networks with ReLU activation functions [26,37].

One such aspect of dense Hopfield networks that is still poorly understood is their performance as generative models for unsupervised learning, where they are trained over some given dataset to reproduce its probability distribution. As far as we are aware, this problem has not yet been studied theoretically for $p$-body models with $p \geq 3$. However, it was studied for the original 2-body Hopfield network by using the teacher-student setting [38] first described in [16,17,39]. In the teacher-student setting, which is also called inverse problem in opposition to the direct problem of random pattern retrieval, a student model $H[\xi|\sigma]$ is trained with $M$ teacher examples $\sigma^a \sim H[\sigma^a|\xi^*]$ conditioned on the planted pattern $\xi^*$. In other words, the student tries to infer the pattern $\xi^*$ of the teacher using a structured set of examples $\sigma^a$.

At finite load $\alpha = \frac{M}{N}$, two regimes of pattern retrieval were found: example retrieval (*eR*) and signal retrieval (*sR*). In the *eR* phase, the student tries to reconstruct $\xi^*$ by directly retrieving the examples $\sigma^a$, which is a good strategy provided that they are strongly correlated with $\xi^*$. In the *sR* phase, on the other hand, retrieval is done by extracting subtle cues from weakly correlated examples. The two types of examples used in these two retrieval strategies are respectively called prototypes and features of $\xi^*$ [26]. Interestingly, a prototype regime and a feature regime were also observed by K & H in dense Hopfield networks trained to classify real data [26], where it was found that the prototype regime is significantly more adversarially robust than the feature regime. In other words, the prototype regime is more resistant than the feature regime to small data perturbations that are specifically designed to cause incorrect classification [40,41]. This prototype approach is arguably a big step towards designing adversarially robust neural networks, a long-standing problem that still lacks a fully satisfying solution [42–44].

In this work, we study the performance of $p$-body Hopfield networks in the teacher-student setting, revealing a prototype regime and a feature regime as in the 2-body model. In Section 2, we review Gardner's main results in studying $p$-body Hopfield models and summarize what the rest of the literature on spin-glass models with $p$-body interactions tell us about the paramagnetic to spin-glass phase transition in $p$-body Hopfield models. In Section 3, we compute the phase diagram of these $p$-body models in the teacher-student setting. In Section 4.1, we discuss the transition to the retrieval phase in the inverse problem. In Section 4.2, we compare this retrieval transition against the transition to the spin-glass phase in the direct problem. Despite their different nature, we show that these two transitions are equivalent on the Nishimori line where the teacher and the student have the same $p$ and $T$ [45–48]. In Section 4.3, we discuss the phase diagram on the Nishimori line in more details. In Section 4.4 and Section 4.5, we discuss the phase diagram outside of the Nishimori line. First of all, we investigate the effect of using an inference temperature different from the dataset noise. Second of all, we reveal that using a larger $p$ for the student than the teacher gives the student an extensive tolerance against both teacher noise and pattern interference. Finally, in Section 4.6, we derive a closed-form expression that measures the adversarial robustness of the student at zero temperature and explain what our results reveal about the nature of adversarial attacks.

## 2  Overview of Gardner's results

Consider the $p$-body Hamiltonian

$$H[\sigma|\xi] = -\sum_{i_1 < ... < i_p = 1}^{N} J_{i_1...i_p} \sigma_{i_1}...\sigma_{i_p} = -\frac{p!}{N^{p-1}} \sum_{i_1 < ... < i_p = 1}^{N} \sum_{\mu=1}^{M} \xi_{i_1}^{\mu}...\xi_{i_p}^{\mu} \sigma_{i_1}...\sigma_{i_p}, \qquad (1)$$

conditioned on a set of $M = \frac{\alpha N^{p-1}}{p!}$ quenched memories $\xi^{\mu} \in \{-1,1\}^N$, $\mu = 1,...,M$, sampled i.i.d. from the Rademacher distribution $\frac{1}{2}\left[\delta\left(\xi_i^{\mu} - 1\right) + \delta\left(\xi_i^{\mu} + 1\right)\right]$. In the *direct model*, pat-

terns $\sigma$ are in turn sampled from the equilibrium Gibbs distribution $P(\sigma|\xi) = Z^{-1}e^{-\beta H[\sigma|\xi]}$, where $\beta \geq 0$ is the inverse temperature and $Z = \sum_\sigma e^{-\beta H[\sigma|\xi]}$ is the system's partition function. The so-called *direct problem* studied by Gardner [30] consists of quantifying the performance of this model as a method of memory retrieval. In that context, the overlap $\frac{1}{N}\sum_i \xi_i^\mu \sigma_i$ is a good measure of retrieval accuracy, and its expected value can be derived from the quenched free entropy $f = \frac{1}{N}\langle \log Z \rangle_\xi$ in the thermodynamic limit $N \to \infty$. At finite $p$, Gardner used the (non-rigorous) replica trick [49] to evaluate the RS approximation of $f$ (see also Appendix B) in terms of a variational principle of the form

$$f = \lim_{N \to \infty} \frac{1}{N}\langle \log Z \rangle_\xi = \lim_{N \to \infty, L \to 0}\left(\frac{\partial}{\partial L}\left[\frac{1}{N}\log\langle Z^L \rangle_\xi\right]\right) = \operatorname*{Extr}_{m,k,q,k,r} f(m,k,q,r),$$

whose solution is

$$
\begin{aligned}
q &= \int_{\mathbb{R}} dx \frac{1}{\sqrt{2\pi}}\exp\left(-\frac{1}{2}x^2\right)\tanh^2\left(\beta\left[\sqrt{\alpha r}x + k\right]\right), \\
m &= \int_{\mathbb{R}} dx \frac{1}{\sqrt{2\pi}}\exp\left(-\frac{1}{2}x^2\right)\tanh\left(\beta\left[\sqrt{\alpha r}x + k\right]\right), \\
r &= pq^{p-1}, \\
k &= pm^{p-1},
\end{aligned}
\tag{2}
$$

and the order parameters $m$ and $q$ are to be interpreted as expected overlaps. To be more precise, $m$ can be shown to be the expected overlap of a retrieval attempt $\sigma$ against one memory in the thermodynamic limit, i.e. $m = \lim_{N \to \infty}\left\langle\frac{1}{N}\sum_i \xi_i^\mu \sigma_i\right\rangle_{\xi,\sigma}$. Similarly, $q$ is the expected overlap between two retrieval attempts $\sigma^1$ and $\sigma^2$, i.e. $q = \lim_{N \to \infty}\left\langle\frac{1}{N}\sum_i \sigma_i^1 \sigma_i^2\right\rangle_{\xi,\sigma}$ or equivalently $q = \lim_{N \to \infty}\left\langle\frac{1}{N}\sum_i \langle\sigma_i\rangle_\sigma^2\right\rangle_\xi$. Intuitively, $q$ measures the tendency of the system to stay frozen in specific configurations rather than visiting all possible values of $\sigma$.

The resulting RS phase diagram (see Fig. 1) are derived from the value of the order parameters as a function of three *hyperparameters*: the interaction order $p$, temperature $T = 1/\beta$ and load $\alpha = \frac{Mp!}{N^{p-1}}$. There are four different phases:

- In the Paramagnetic phase ($P$), the overlaps $m$ and $q$ both vanish. The network does not retrieve any specific pattern: sampled configurations are completely random.

- In the Spin-Glass phase ($SG$), $m$ vanishes but $q > 0$. In other terms, the network does not retrieve individual stored memories but rather converges to spurious patterns depending on all the memories in a non-trivial way.

- In the signal Retrieval phases ($lR$ and $gR$), $m \neq 0$ and $q > 0$, which means that the network is able to retrieve the stored memories. $lR$ and $gR$ are respectively locally stable and globally stable. In other words, local retrieval $lR$ is only attainable from initial conditions in a limited neighborhood of a memory $\xi^\mu$, while global retrieval $gR$ is accessible from any initial conditions given enough time. These two phases are said to be ferromagnetic.

Gardner also calculated the exact $p \to \infty$ phase diagram without making any assumptions about replica symmetry [30]. In this limit, the resulting paramagnetic to spin-glass ($P$-$SG$) phase transition occurs at a temperature $T_E(\alpha)$ that coincides with the boundary of the region where the total entropy of the paramagnetic phase becomes negative, given by $\beta^2 \alpha = 2\log 2$ (white dashed line in Fig. 1). At finite $p$, Gardner's results only tell us that the model cannot be in the paramagnetic phase below $T_E(\alpha)$. Therefore, a spin-glass transition should occur at a temperature $T_s(\alpha,p) \geq T_E(\alpha)$.

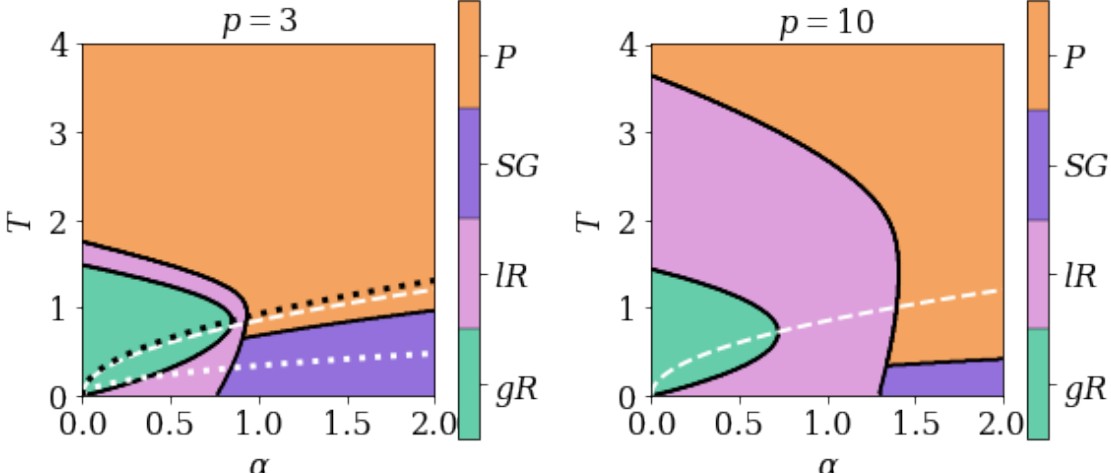

Figure 1: RS phase diagrams of the direct models with $p = 3$ on the left and $p = 10$ on the right. Accurate pattern retrieval is not possible in the paramagnetic phase ($P$) or in the spin-glass phase ($SG$), but it is possible in the local retrieval phase ($lR$) and in the global retrieval phase ($gR$). The ferromagnetic fixed point corresponding to accurate pattern retrieval is globally stable in the $gR$ phase, but locally stable in the $lR$ phase. The phase diagrams are inexact below the white dashed line where the total entropy of the paramagnetic phase becomes negative. The black dotted line overlaying the $p = 3$ diagram is the (exact) 1RSB $P$-$SG$ transition temperature $T_s(\alpha, 3)$, which is obtained by rescaling by $\sqrt{2\alpha}$ the corresponding transition temperature of the spin-glass model with $p$-body Gaussian interactions. The d1RSB transition $T_d(\alpha, 3)$ is very close to $T_s(\alpha, 3)$ throughout the displayed range of $\alpha$. The white dotted line in the $p = 3$ plot is the temperature $T_G(\alpha, 3)$ below which multiple steps of RSB are required to compute the free entropy. It is also obtained by rescaling by $\sqrt{2\alpha}$ the corresponding transition temperature of the Gaussian spin-glass model.

Since the RS spin glass solution of Eqs. (2) exists only below $T_E(\alpha)$ (violet region in Fig. 1), the spin-glass transition must be towards a RSB spin-glass phase.

Outside of the signal retrieval phases, the free entropy of the direct model is the same as for the spin-glass model with $p$-body Gaussian interactions where the temperature is rescaled by a factor of $\sqrt{2\alpha}$ [50, 51]. Therefore, the spin-glass and paramagnetic solutions are the same in the direct model as in this Gaussian spin-glass model, and we expect the exact phase diagrams of both models to be identical when the direct model is not in its signal retrieval phases. According to previous work on the Gaussian model with finite $p$ [51], a 1RSB solution with $m = k = 0$ exists and is globally stable throughout a whole phase below $T_s(\alpha, p) \geq T_E(\alpha)$ (see Fig. 1). This solution becomes unstable at a lower transition temperature $T_G(\alpha, p)$ (see Fig. 2), below which multiple steps of RSB are required. In the limit of $p \to \infty$, it holds that $T_s(\alpha, p) \to T_E(\alpha)$ and $T_G(\alpha, p) \to 0$. In other terms, the direct model becomes 1RSB, which is consistent with the fact that it is converging to a random energy model with temperature rescaled by $\sqrt{2\alpha}$ [30, 50, 52]. Finally, we mention that this type of models exhibits a random first order transition phenomenology [53–56]: there is in fact a range of temperatures $T_s(\alpha, p) \leq T \leq T_d(\alpha, p)$ where the dynamics get trapped in an exponential number of metastable clusters, with an emerging RSB structure that does not affect the free energy (see Fig. 2). This range of temperatures thus defines a so-called dynamical 1RSB (d1RSB) phase. Below $T_s(\alpha, p)$, the number of clusters is no longer exponential, and the system undergoes the thermodynamic 1RSB phase transition that we mentioned previously. The critical

temperatures $T_G(\alpha, p)$, $T_s(\alpha, p)$ and $T_d(\alpha, p)$ can all be obtained by standard RSB methods, but the resulting saddle-point equations can be prone to numerical instability at large $p$ [33]. In Sections 4.2 and 4.3, we discuss an alternative way to obtain $T_s(\alpha, p)$ and $T_d(\alpha, p)$.

## 3 Teacher-student setting

On our end, we study a dense Hopfield network with Hamiltonian (1) as a generative model for unsupervised learning. In that context, the memories $\xi$ are model parameters that have to be trained in such a way that the examples of a given dataset $\{\sigma^a\}_{a=1}^M$ result as typical network configurations.

In particular, we study a controlled teacher-student setting in which the examples are sampled from the probability distribution $P(\sigma^a|\xi^*)$ of a so-called *teacher* dense Hopfield network conditioned on a single *planted* pattern $\xi^* \in \{-1, 1\}^N$ whose entries are quenched Rademacher random variables. A *student* dense Hopfield network, also known as the *inverse model*, then samples its own student pattern $\xi$ from the posterior distribution

$$P(\xi|\sigma) = \frac{P(\xi)\prod_{a=1}^M P(\sigma^a|\xi)}{P(\sigma)} = \frac{P(\xi)}{P(\sigma)}\prod_{a=1}^M Z^{-1}\exp\left(-\beta H[\sigma^a|\xi]\right),$$

where $P(\sigma^a|\xi)$ is the Gibbs distribution of the direct model with a single memory $\xi$, and $P(\xi)$ is the prior on $\xi$ that is chosen to be uniform. Since the direct model has only a single pattern, $Z$ does not depend on $\xi$ (see Appendix C), and the posterior simplifies to

$$P(\xi|\sigma) = \mathcal{Z}^{-1}(\sigma)\exp\left(-\beta H[\xi|\sigma]\right).$$

In sum, the student posterior distribution is that of a dense Hopfield network where $\xi$ plays the role of the sampled pattern and the examples $\sigma$ act like the $M$ quenched memories. Our task, called the *inverse problem*, consists of quantifying the student's capability to infer the teacher pattern, which we will also call the *signal*. Like Gardner, we calculate a free entropy of the form $f = \frac{1}{N}\langle \log \mathcal{Z} \rangle_\sigma$ in the thermodynamic limit $N \to \infty$. This time, however, the average $\langle \cdot \rangle_\sigma$ is over a structured set of examples $\sigma$. In fact, we recall that, unlike the i.i.d. memories studied by Gardner, the examples $\sigma^a$ are sampled from the teacher distribution $P(\sigma^a|\xi^*)$.

In general, the student does not have access to the teacher generative model. In our controlled teacher-student setting, the student knows that the correct model for $P(\sigma^a|\xi)$ is a dense Hopfield network. Nevertheless, it does not necessarily have access to the interaction order $p^*$ and inverse temperature $\beta^*$ used by the teacher. Therefore, we denote the student hyperparameters by $p$ and $\beta$ and emphasize that they are not necessarily equal to $p^*$ and $\beta^*$. As previously stated, we calculate the free entropy

$$f = \frac{1}{N}\langle \log \mathcal{Z} \rangle_\sigma = 2^{-N}\sum_{\xi^*}\sum_\sigma [Z^*]^{-M}\exp\left(\beta^*\frac{p^*!}{N^{p^*-1}}\sum_{a=1}^M\sum_{i_1<...<i_{p^*}}\xi_{i_1}^*...\xi_{i_{p^*}}^*\sigma_{i_1}^a...\sigma_{i_p}^a\right)$$
$$\times \log\sum_\xi \exp\left(\beta\frac{p!}{N^{p-1}}\sum_{a=1}^M\sum_{i_1<...<i_p}\xi_{i_1}^b...\xi_{i_p}^b\sigma_{i_1}^a...\sigma_{i_p}^a\right), \tag{3}$$

in the thermodynamic limit $N \to \infty$. We then draw phase diagrams of the inverse problem as a function of $p^*$, $T^* = 1/\beta^*$, $p$, $T = 1/\beta$ and $\alpha$, where $\alpha$ is $M$ normalized to $\mathcal{O}(1)$. Unless explicitly specified otherwise, we use $\alpha = \frac{Mp!}{N^{p-1}}$.

### 3.1 Matched interaction orders

We first consider the case where $p^* = p$ and the only possible mismatch between the teacher and student networks is in the inverse temperature, i.e. $\beta^* \neq \beta$. At low $T^*$, the student's task is easy. In fact, below the critical temperature $T_{\text{crit}}$ of the direct problem with one pattern (see Fig. 1, $\alpha = 0$ axis), the teacher produces examples $\sigma^a$ that cluster around $\xi^*$. Therefore, the student can infer $\xi^*$ by aligning its pattern $\xi$ with the examples $\sigma^a$. This retrieval strategy works even when using a very small amount of examples (see [38]). Since the size of our dataset is extensive, the retrieval accuracy is maximum in the thermodynamic limit. We call this region the (accurate) example Retrieval phase ($eR$).

Conversely, when $T^*$ is above $T_{\text{crit}}$, the examples in the training set are very noisy and we do not observe a finite overlap between $\sigma^a$ and $\xi^*$ (see Fig. 1, $\alpha = 0$ axis). In this regime, we find that the RS approximation of the $p^* = p$ free entropy can be computed (see Appendix D) in terms of the variational principle

$$
f = \underset{m,k,q,r,q^*,r^*}{\text{Extr}} \left\{ \beta^* \beta \alpha [q^*]^p - \frac{1}{2} \beta^2 \alpha q^p + \beta m^p - \beta^* \beta \alpha r^* q^* \right.
$$
$$
+ \frac{1}{2} \beta^2 \alpha r q - \frac{1}{2} \beta^2 \alpha r - \beta m k + \frac{1}{2} \beta^2 \alpha + \log 2
$$
$$
\left. + \int dx \frac{1}{\sqrt{2\pi}} \exp\left\{ -\frac{1}{2} x^2 \right\} \left\langle \log\left[ \cosh\left( \beta \left[ \sqrt{\alpha r} x + \beta^* \alpha r^* + kz \right] \right) \right] \right\rangle_z \right\}, \tag{4}
$$

whose solution is the saddle-point equations

$$
q^* = \int_{\mathbb{R}} dx \frac{1}{\sqrt{2\pi}} \exp\left( -\frac{1}{2} x^2 \right) \left\langle \tanh\left( \beta \left[ \sqrt{\alpha r} x + \beta^* \alpha r^* + kz \right] \right) \right\rangle_z,
$$
$$
q = \int_{\mathbb{R}} dx \frac{1}{\sqrt{2\pi}} \exp\left( -\frac{1}{2} x^2 \right) \left\langle \tanh^2\left( \beta \left[ \sqrt{\alpha r} x + \beta^* \alpha r^* + kz \right] \right) \right\rangle_z,
$$
$$
m = \int_{\mathbb{R}} dx \frac{1}{\sqrt{2\pi}} \exp\left( -\frac{1}{2} x^2 \right) \left\langle z \tanh\left( \beta \left[ \sqrt{\alpha r} x + \beta^* \alpha r^* + kz \right] \right) \right\rangle_z, \tag{5}
$$
$$
r^* = p [q^*]^{p-1},
$$
$$
r = p q^{p-1},
$$
$$
k = p m^{p-1},
$$

where $z$ is a Rademacher random variable and $\alpha = \frac{Mp!}{N^{p-1}}$. As in the direct model described in Section 2, the order parameters $m$ and $q$ have a clear interpretation in terms of expected overlaps. $m = \lim_{N\to\infty} \left\langle \frac{1}{N} \sum_i \xi_i \sigma_i^a \right\rangle_{\xi^*,\sigma,\xi}$ is the expected overlap of a retrieval attempt with an example $\sigma^a$, and $q = \lim_{N\to\infty} \left\langle \frac{1}{N} \sum_i \langle \xi_i \rangle_\xi^2 \right\rangle_{\xi^*,\sigma}$ is the expected overlap between two retrieval attempts. Similarly, $q^*$ is the expected overlap between the teacher and student patterns, i.e. $q^* = \lim_{N\to\infty} \left\langle \frac{1}{N} \sum_i \xi_i^* \xi_i \right\rangle_{\xi^*,\sigma,\xi}$. Therefore, it is a good measure of inference performance. The free entropy (Eq. 4) is expected to be exact in absence of mismatch between the teacher and the student, i.e. $\beta^* = \beta$. This condition is known as the Nishimori line [45–48]. Outside of the Nishimori region, RSB corrections are expected. Like the direct problem, the inverse problem with $T^* > T_{\text{crit}}$ has different phases characterized by the values of the order parameters:

- In the Paramagnetic phase ($P$), the overlaps $m$, $q^*$ and $q$ all vanish.

- In the signal Retrieval phases ($lR$ and $gR$), $m = 0$ but $q^* \neq 0$ and $q > 0$. $lR$ and $gR$ are respectively locally stable and globally stable. In other words, local retrieval $lR$ is only

attainable from initial conditions in a limited neighborhood of $\xi^*$, while global retrieval $gR$ is accessible from any initial conditions given enough time. These two phases are also said to be ferromagnetic.

- In the (inaccurate) example Retrieval phase ($eR$), $m \neq 0$ and $q > 0$ but $q^* = 0$.

- In the Spin-Glass phase (SG), $q > 0$ but $q^*$ and $m$ vanish.

In sum, when $T^*$ is above $T_{\text{crit}}$, the student can only learn the teacher pattern in the signal retrieval phases. In all the other phases, the student pattern is uncorrelated with the signal, being either a random guess ($P$ phase), aligned with a noisy example (inaccurate $eR$ phase), or aligned with a spurious low energy state ($SG$ phase). We stress that we cannot have $m \neq 0$ and $q^* \neq 0$ at the same time (accurate $eR$ phase) when $T^* > T_{\text{crit}}$ because $\lim_{N \to \infty} \left\langle \frac{1}{N} \sum_i \xi_i^* \sigma_i^a \right\rangle_{\xi^*, \sigma} = 0$ in that regime (see Fig. 1, $\alpha = 0$ axis).

## 3.2 Mismatched interaction orders

We also investigate the $T^* > T_{\text{crit}}$ regime in the presence of a mismatch between the interaction orders of the teacher and student networks, i.e. $p^* \neq p$. We focus on the case of $p^* = 2$ and even $p \geq 3$ to study the consequences of fitting the teacher of [38] using a student with higher order interactions. We find two different scaling regimes of the training set size $M$ and inverse temperature $\beta^*$ that make retrieval possible (see Appendix D):

- a large-noise scaling where $\beta^* \sim \mathcal{O}\left(N^{2/p-1}\right)$ and $M \sim \mathcal{O}\left(N^{p-1}\right)$, such that $\alpha = \frac{Mp!}{N^{p-1}}$ and $\lambda = \frac{[\beta^*]^{p/2}}{(p/2)!} N^{p/2-1}$ are finite;

- a finite-noise scaling where $\beta^* \sim \mathcal{O}(1)$ and $M \sim \mathcal{O}\left(N^{p/2}\right)$, such that $\alpha = \frac{M(p/2+1)!}{N^{p/2}}$ is finite.

In the large-noise scaling, we obtain saddle point equations similar to Eqs. (5) but with $\beta^*$ replaced by $\lambda$ (see Appendix D). Conversely, the finite noise scaling leads to

$$
\begin{aligned}
q^* &= \left\langle \tanh\left(\beta \left[\eta \alpha r^* + kz\right]\right) \right\rangle_z, \\
m &= \left\langle z \tanh\left(\beta \left[\eta \alpha r^* + kz\right]\right) \right\rangle_z, \\
r^* &= p \left[q^*\right]^{p-1}, \\
k &= p m^{p-1},
\end{aligned}
\tag{6}
$$

where $\eta$ generally depends on $\beta^*$ and $p$ in a non-trivial way, but we find that $\eta = \frac{2[\beta^*]^2}{(1-2\beta^*)^2}$ when $p = 4$ (see Appendix D). These equations can also be derived by extrapolating the large-noise equations to $\alpha_{\text{large noise}} \to 0$ and $\lambda \to \infty$ with fixed $\lambda \alpha_{\text{large noise}} = \eta \alpha_{\text{finite noise}}$.

# 4 Results and Discussion

## 4.1 Retrieval transition at large interaction order

The paramagnetic solution of Eqs. (5) always exists and is globally stable in the part of the phase diagram where the temperature $T$ is relatively large and $\alpha = \frac{Mp!}{N^{p-1}}$ is relatively small. On the other hand, the $gR$ phase exists when $\beta^2 \alpha p$ and $\beta^* \beta \alpha p$ are both large. In fact, in that limit, $q^* = q = 1$ is a fixed point of Eqs. (5). The critical line where $gR$ becomes globally stable instead of $P$ is not clear from this analysis alone, but we can at least find it analytically in

the limit of infinite $p$. As for the direct model, the free entropy and the total entropy of the paramagnetic phase are respectively $\frac{1}{2}\beta^2\alpha + \log 2$ and $-\frac{1}{2}\beta^2\alpha + \log 2$ [30]. At the same time, the $p \to \infty$ free entropy takes the form

$$
\begin{aligned}
f = \mathrm{Extr}\Big\{ &\beta^*\beta\alpha\,\theta\,(q^*-1) - \frac{1}{2}\beta^2\alpha\,\theta\,(q-1) - \beta^*\beta\alpha r^* q^* + \frac{1}{2}\beta^2\alpha r q - \frac{1}{2}\beta^2\alpha r + \frac{1}{2}\beta^2\alpha \\
&+ \log 2 + \int dx \frac{1}{\sqrt{2\pi}}\exp\Big\{-\frac{1}{2}x^2\Big\}\log\Big[\cosh\Big(\sqrt{\beta^2\alpha r}\,x + \beta^*\beta\alpha r^*\Big)\Big]\Big\},
\end{aligned}
$$

where $\theta\,(q-1) := \lim_{p\to\infty} q^p$, $q \in [0,1]$, is the Heaviside step function jumping at $q = 1$, i.e. $\theta(1) = 1$ and $\theta(q) = 0 \; \forall q \in [0,1)$. In this limit, the ferromagnetic phase is characterized by $q = q^* = 1$, and its free entropy is then

$$
\begin{aligned}
f &= \beta^*\beta\alpha - \beta^*\beta\alpha p + \int dx \frac{1}{\sqrt{2\pi}}\exp\Big\{-\frac{1}{2}x^2\Big\}\log\Big[2\cosh\Big(\sqrt{\beta^2\alpha p}\,x + \beta^*\beta\alpha p\Big)\Big] \\
&\approx \beta^*\beta\alpha - \beta^*\beta\alpha p + \int dx \frac{1}{\sqrt{2\pi}}\exp\Big\{-\frac{1}{2}x^2\Big\}\Big(\sqrt{\beta^2\alpha p}\,x + \beta^*\beta\alpha p\Big) \\
&= \beta^*\beta\alpha.
\end{aligned}
$$

The corresponding total entropy is $s = f - \beta\frac{\partial f}{\partial \beta} = 0$, as expected from a ferromagnetic phase with $q^* = q = 1$. On the Nishimori line, $f = \beta^*\beta\alpha$ becomes larger than the free entropy of the paramagnetic phase, which triggers a phase transition, if and only if

$$
T < \sqrt{\frac{\alpha}{2\log 2}}, \tag{7}
$$

where $T_E = \sqrt{\frac{\alpha}{2\log 2}}$ is also the temperature below which the total entropy of the paramagnetic phase becomes negative. Outside of the Nishimori line, this inequality generalizes to $\beta^*\beta\alpha > \frac{1}{2}\beta^2\alpha + \log 2$, leading to

$$
\beta^* - \sqrt{[\beta^*]^2 - \frac{2\log 2}{\alpha}} < \beta < \beta^* + \sqrt{[\beta^*]^2 - \frac{2\log 2}{\alpha}},
$$

while the temperature where the paramagnetic total entropy becomes negative stays the same.

## 4.2  Transition to the ordered phases: Universality

In the $p \to \infty$ limit, the transition towards $gR$ of the inverse model on the Nishimori line is identical to the exact $P$-$SG$ transition of the direct model [30]. We claim that these two critical lines are actually closely related for any $p$. In the Hopfield model with $p = 2$, they were already shown to be identical [38]. We will now argue that they overlap for any $p$ and $\beta$ such that $T > T_{\mathrm{crit}}$ (see Figs. 2 and 1). In the case of $p = 2$, both lines can be obtained exactly from the RS approximation of either the direct model or the inverse model, so there is no obvious advantage to using this equivalence in calculations. In general, while the inverse problem on the Nishimori line is replica symmetric, the direct problem is not, and the $p \geq 3$ replica symmetric $P$-$SG$ transition is not exact. Moreover, even the critical line calculated using 1RSB may be inaccurate due to numerical instability [33]. In this situation, the knowledge of the $gR$ transition in the replica-symmetric inverse problem can be used to locate the exact $P$-$SG$ transition of the direct problem, where symmetry breaking occurs.

For that purpose, we will argue that, given $T > T_{\mathrm{crit}}$, *the direct model is in the paramagnetic phase if and only if the inverse model is in the paramagnetic phase.*

The converse implication comes from the fact that since (see Appendix C)

$$P(\sigma) = \frac{1}{2^{MN}} \frac{\mathcal{Z}(\sigma)}{\langle \mathcal{Z} \rangle}, \tag{8}$$

the example distribution $P(\sigma)$ of the inverse problem is contiguous [57] to the uniform distribution, i.e. the memory distribution of the direct problem, when

$$\lim_{N \to \infty} \left\{ \frac{\log \mathcal{Z} - \log \langle \mathcal{Z} \rangle}{N} \right\} = 0. \tag{9}$$

As determined in Appendix C and D, the annealed expression $\frac{1}{N} \log \langle \mathcal{Z} \rangle$ is equal to the free entropy of the paramagnetic phase. Therefore, when the inverse model is in the paramagnetic phase, $P(\sigma)$ is contiguous to the uniform distribution. This property is called quiet planting and is known to occur more generally in mean-field paramagnets [58–61]. In our problem setting, it means that if the inverse model is in the paramagnetic phase, then it is equivalent to the direct model. In particular, if the inverse model is in the paramagnetic phase, then so is the direct model. In more intuitive terms, the *gR* transition temperature of the inverse model must be greater than or equal to the *P-SG* transition temperature of the direct model because the ensemble of examples $\sigma^a$ generated by the teacher model is on average at least as structured as the set of i.i.d. random memories stored in the direct model.

For the direct implication, notice that the average replicated partition function of the direct model in the paramagnetic phase can be approximated as (see Appendix E)

$$\langle Z^L \rangle \approx \frac{1}{\langle Z \rangle} \left\langle \sum_{\sigma} \exp\left( \beta N \sum_{\gamma} \sum_{\mu \in \Gamma_{\gamma}} \left[ \frac{1}{N} \sum_{i} \xi_i^{\mu} \sigma_i^{\gamma} \right]^p \right.\right.$$
$$\left. + \beta \sum_{\gamma} \sum_{\mu \in \bar{\Gamma}} \frac{p!}{N^{p-1}} \sum_{i_1 < ... < i_p} \xi_{i_1}^{\mu} ... \xi_{i_p}^{\mu} \sigma_{i_1}^{\gamma} ... \sigma_{i_p}^{\gamma} \right)$$
$$\left. \sum_{\sigma_0} \exp\left( \beta \sum_{\mu \in \bar{\Gamma}} \frac{p!}{N^{p-1}} \sum_{i_1 < ... < i_p} \xi_{i_1}^{\mu} ... \xi_{i_p}^{\mu} \sigma_{i_1}^{0} ... \sigma_{i_p}^{0} \right) \right\rangle.$$

This expression is identical to the replicated partition function of the inverse model with $T > T_{\text{crit}}$, which therefore must also be in the paramagnetic phase.

As a consequence, when $T > T_{\text{crit}}$, the *P-SG* transition line of the direct model must be identical to the *gR* transition line of the inverse model on the Nishimori line.

### 4.3 Phase diagram on the Nishimori line

On the Nishimori line, the student is fully informed about the teacher generative model and uses $\beta = \beta^*$ and $p = p^*$. In this scenario, thanks to the Nishimori identities [46], it is well known that $\xi^*$ and $\xi$ play symmetric roles and that $q^* = q$. For the same reason, the overlaps $\frac{1}{N} \sum_i \xi_i^* \xi_i$ and $\frac{1}{N} \sum_i \xi_i^1 \xi_i^2$ have the same distribution. From the self-averaging of $\frac{1}{N} \sum_i \xi_i^* \xi_i$, it follows that the system is expected to be replica symmetric, and Eqs. (4) and (5) are expected to hold. Fig. (2) shows the phase diagrams obtained by solving the saddle-point equations numerically on the Nishimori line. Both $q^* = q$ and the replica symmetry condition are verified. In particular, numerical solutions of a few values of $p \geq 3$ show that the *gR* transition occurs at a higher $T$ than the line $\beta^2 \alpha = 2 \log 2$ where the total entropy of the paramagnetic phase becomes negative. In other terms, the phase transition towards *gR* prevents the total entropy from becoming negative when $T$ decreases below $\sqrt{\frac{\alpha}{2 \log 2}}$, which is consistent with the RS solution being exact on the Nishimori line.

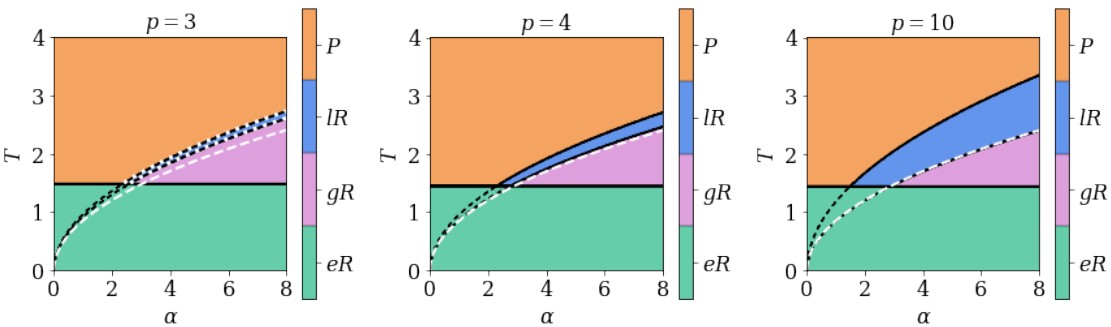

Figure 2: Exact RS phase diagrams of inverse models on the Nishimori line, i.e. $p^* = p$ and $\beta^* = \beta$. the left, center and right plots respectively have $p = 3$, $p = 4$ and $p = 10$. Accurate pattern retrieval is not possible in the paramagnetic phase ($P$), but it is possible in the local retrieval phase ($lR$), in the global retrieval phase ($gR$) and in the example retrieval phase ($eR$). The ferromagnetic fixed point corresponding to accurate pattern retrieval is globally stable in the $gR$ phase, but locally stable in the $lR$ phase. The critical temperature of the $eR$ phase is the critical temperature $T_{\text{crit}}$ of the direct problem with one pattern (see Fig. 1, $\alpha = 0$ axis). The black dashed lines mark the spurious continuation of the $lR$ and $gR$ phase boundaries through the $eR$ phase. The white dashed line is the $p \to \infty$ $gR$ critical line calculated analytically in Section 4.1. It matches the corresponding numerical phase boundary increasingly well as $p$ grows larger. The white dotted lines on the $p = 3$ plot mark the 1RSB and d1RSB critical temperatures $T_s(\alpha, 3)$ and $T_d(\alpha, 3)$ of the direct model (see Section 2). We truncated them below $T_{\text{crit}}$ for improved visibility. $T_s(\alpha, 3)$ and $T_d(\alpha, 3)$ are obtained by rescaling the corresponding critical temperatures found in [54] by $\sqrt{2\alpha}$.

At low $T$, the student can learn efficiently within the accurate $eR$ regime. In this phase, learning is possible ($q^* \neq 0$) because the examples are correlated with the signal and the student can retrieve it by simply being aligned with them ($m \neq 0$).

At high $T$, learning is possible only if the amount of examples, i.e. the size of the dataset, is sufficiently large. When $\alpha$ is too small, Eqs. (5) have only a paramagnetic fixed point because the amount of information carried by the dataset is not large enough. Numerical solutions suggest that the paramagnetic fixed point always exist and it is actually locally stable in the whole high-temperature regime. When $\alpha$ is sufficiently large, the signal retrieval fixed point appears as a locally stable attractor ($lR$ phase). It becomes globally stable ($gR$ phase) as the size of the dataset is increased further or the student temperature decreases.

As per the previous Section, the critical boundary of the $gR$ phase obtained by solving Eqs. 5 is identical to the 1RSB $P$-$SG$ transition temperature $T_s(\alpha, p)$ of the direct model. Similarly, we observe that the metastable $lR$ phase coincides with the d1RSB phase of the direct model (see Fig. 2). Our results are also consistent with the fact that $T_s(\alpha, p) \to T_E(\alpha)$ in the $p \to \infty$ limit. In fact, we find that the analytical limit boundary closely agrees with the numerical solution of the saddle-point equations with $p^* = p = 10$ and remains a good approximation even down to $p^* = p = 4$.

In the student model, $\sigma$ plays a similar role as the weights of the trainable dense Hopfield network model that K & H designed for classification of data [26]. In that context, $\xi$ is analogous to the test data whose labels are being predicted (see Fig. 3). In fact, the computation performed by K & H's model to recover labels is similar to the update rule used by the student to infer the teacher pattern (see Appendix A). Moreover, the $eR$ and $gR$ phases are respectively reminiscent of the prototype and feature regimes of K & H's networks. Therefore, we believe that the student can act as a toy model of label prediction in these two regimes.

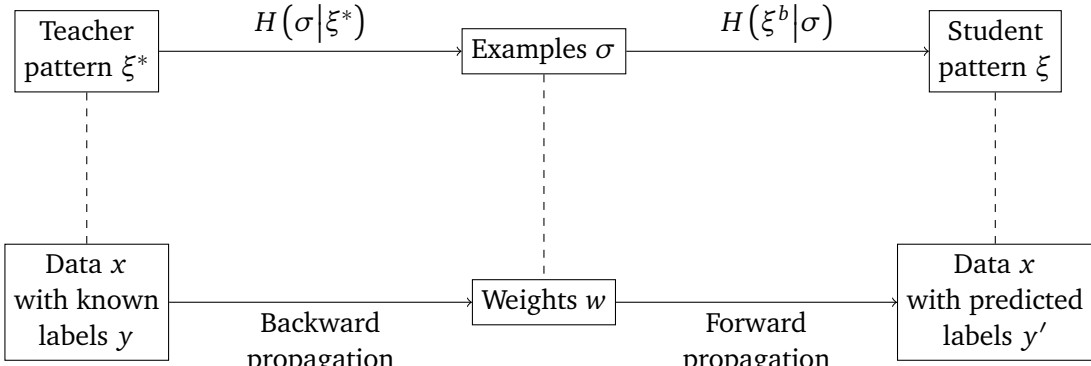

Figure 3: The first row of this diagram sketches how a $p$-body Hopfield network in the teacher-student setting can reconstruct an incomplete pattern $\xi^b$ to match the teacher pattern $\xi^*$ by relying on the examples $\sigma$ obtained from $\xi^*$. The second row summarizes how a dense neural network trained by K & H can recover the labels $y'$ of the data $x$ given the weights $w$ learned from $x$ [26]. Both models tackle similar tasks using an approach where $\sigma$ and $\xi^b$ respectively play the same roles as $w$ and $(x, y')$. The forward propagation algorithm used to generate $y'$ is similar to the update rule of the student (see [26] and Appendix A), but the backpropagation algorithm used to learn $w$ is very different from the update rule of the teacher.

Comparing instead the phase diagrams of our inverse model with that of the inverse 2-body Hopfield model, we see that the *eR* and *gR* phases of the inverse $p$-body model with $p \geq 3$ are respectively analogous to the *eR* and *sR* (signal Retrieval) phases presented in [38]. One of the key differences between $p = 2$ and $p \geq 3$ is that the paramagnetic to signal retrieval phase transition of the $p$-body model is second order for $p = 2$ but first order for $p \geq 3$. On the one hand, the second order phase transition of $p = 2$ indicates that its paramagnetic fixed point is never locally stable and sets an unambiguous boundary between the *sR* phase where $\xi^*$ can be recovered starting from any initial conditions and the paramagnetic phase where pattern retrieval is impossible [61]. On the other hand, the first order phase transition of $p \geq 3$ allows the retrieval and paramagnetic regimes to coexist. The *lR* phase is locally stable precisely because it coexists with the paramagnetic phase and has a lower free entropy. Meanwhile, the *gR* phase also coexists with the paramagnetic phase, but has a larger free entropy. In the presence of phase coexistence, an algorithm trying to retrieve $\xi^*$ starting from random initial conditions can get stuck in the paramagnetic phase instead. In fact, it has been conjectured that there is no algorithm with random initial conditions that can find such a ferromagnetic fixed point in a tractable amount of time [61, 62]. That kind of metastable region was thus given the name *hard phase* [61, 63]. In summary, we expect that $p \geq 3$ models in the *gR* phase can only recover partially corrupted patterns whereas $p = 2$ can recover them entirely.

Fig. (4) shows results from Monte Carlo simulations with $p = 3$, where $L$ replicas of the student pattern $\{\xi^b\}_{b=1}^L$ are initialized to the teacher pattern $\xi^*$ corrupted by some Rademacher noise $\varepsilon$. In other words, the initial values of $\xi_i^b$ are sampled from the distribution $(1 - \varepsilon) \delta (\xi_i - \xi_i^*) + \frac{\varepsilon}{2} [\delta (\xi_i + 1) + \delta (\xi_i - 1)]$ with $\varepsilon \in [0, 1]$. The value of $\varepsilon$ is tuned so that the simulations start relatively close to the saddle-point solutions. As explained previously, *gR* is a hard phase, so this initialization is necessary to make $\xi^b$ converge to *gR* in a reasonable amount of time. Additionally, it is also used to make $\xi^b$ converge to the *lR* phase rather than the *P* phase when desired. Once the simulations are over, the overlaps are averaged over all $L$ replicas. If we fix $\varepsilon = 0$, then the simulations generally converge to the *lR* phase when it is a fixed point. If instead we initialize them to the saddle-point solutions by handpicking $\varepsilon$, then

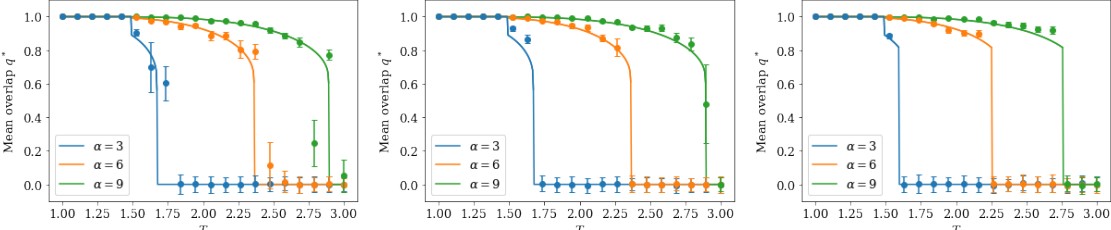

Figure 4: Monte-Carlo simulations of the $p = 3$ inverse model compared against RS saddle-point solutions. The $lR$ phase is included on the left and central plots, but not on the right one. The left plot has $\varepsilon = 0$, and the two other ones have a handpicked $\varepsilon$ such that the simulations are initalized near the saddle-point solutions. The dots are simulation data at a few values of $\alpha$, and the lines are slices of the saddle-point solutions at the same $\alpha$. The teacher generates $M = \frac{\alpha N^{p-1}}{p!}$ examples $\sigma^a$ with $N = 512$ components each, and the simulation results are then averaged over $L = 100$ student patterns. The simulation data is sometimes systematically shifted up with respect to the saddle-point solution. This difference is notably visible on the central plot, right after the fall from $eR$ to $gR$ when $\alpha = 3$.

they stay near the initial overlaps. In either case, the simulations converge to $eR$ when it is globally stable. Some simulation data points might be systematically shifted up with respect to the saddle-point solutions. However, this difference decreases with the system size $N$, so finite size effects seem sufficient to explain it (see Fig. 9 in Appendix F). Overall, the Monte-Carlo simulations are in very good agreement with the $p = 3$ overlap landscape obtained by solving the saddle-point equations numerically.

## 4.4 Inference temperature vs dataset noise

In the two next Sections, we will discuss the phase diagram when the student is only partially informed about the teacher generative model, i.e. when the Nishimori conditions do not hold. We start with the case where $p = p^*$ but $\beta \neq \beta^*$, i.e. the inference temperature $T$ is different from the dataset noise $T^*$. As we argued in Section 3.1, the student accurately retrieves $\xi^*$ when $T^* < T_{\text{crit}}$. On the other hand, we must solve the saddle-points equations (see Eqs. 5) to study $T^* > T_{\text{crit}}$.

We show the phase diagram of this region on Fig. (5). At high inference temperature $T$, the situation is similar to Fig. (2): retrieval is possible if the data load $\alpha$ is sufficiently large, but the paramagnetic phase is always locally stable. The situation is different when the inference temperature is low. In that case, there are two phases that we did not see for $\beta = \beta^*$: the inaccurate $eR$ phase and the $SG$ phase. When $\alpha$ is relatively small, the student falls in the inaccurate $eR$ phase. In this regime, it has finite overlap with one of the noisy examples and cannot retrieve the signal $\xi^*$. When $\alpha$ is larger, the interference among the noisy examples prevents the student to be aligned with them. In this regime, the $SG$ phase, the student locally converge to spurious patterns that are uncorrelated with the signal.

Accurate pattern retrieval is only possible in the $lR$ and $gR$ phases where $\alpha$ is so large that the student can gather enough information from the dataset to become very close to $\xi^*$. The phase diagrams indicate that pattern retrieval is optimal on the Nishimori line in the sense that $\beta = \beta^*$ is the inverse temperature where the student needs the least examples to recover $\xi^*$. In other words, the student's performance is non-monotonic in $T$ and peaks at $T = T^*$. These properties were also observed in the teacher-student setting of the $p = 2$ Hopfield network [38].

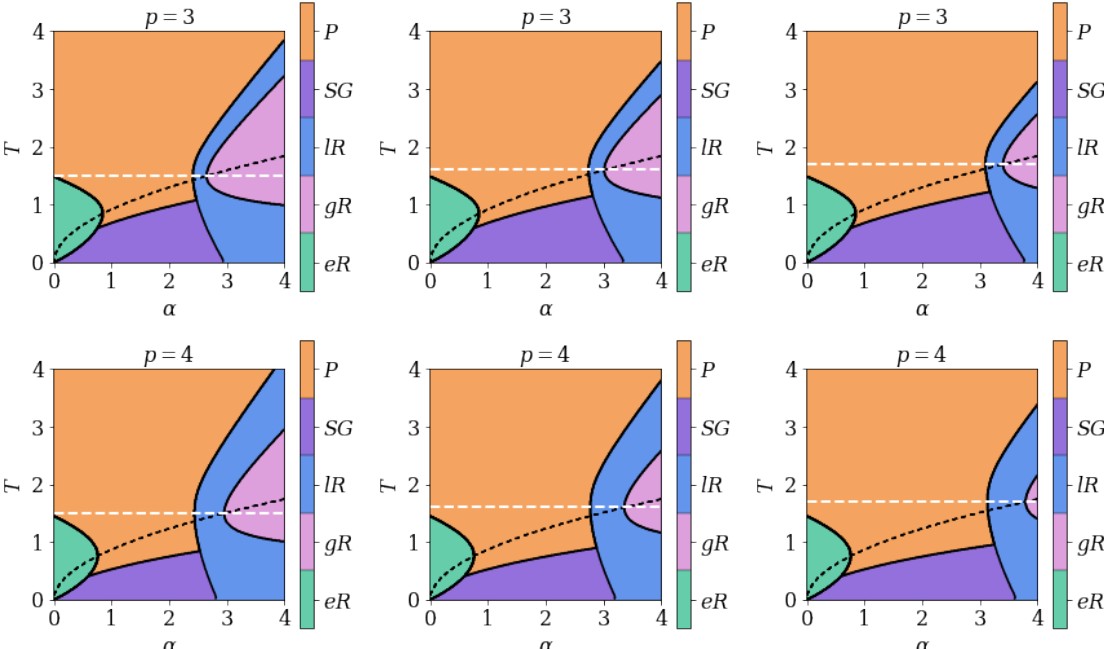

Figure 5: RS phase diagrams of inverse models with $p^* = p$ and fixed $\beta^*$. The top and bottom rows of plots respectively have $p^* = p = 3$ and $p^* = p = 4$. In the same way, the left, central and right columns correspond to $T^* = 1.5$, $T^* = 1.6$ and $T^* = 1.7$. Accurate pattern retrieval is not possible in the paramagnetic phase ($P$), in the spin-glass phase ($SG$) or in the example retrieval phase ($eR$), but it is possible in the local retrieval phase ($lR$) and in the global retrieval phase ($gR$). The ferromagnetic fixed point corresponding to accurate pattern retrieval is globally stable in the $gR$ phase, but locally stable in the $lR$ phase. Conversely, the $SG$ fixed point is always locally stable and leads the student to a frozen spurious signal. The white dashed line indicates the Nishimori line $\beta^* = \beta$. The black dashed lined is the $gR$ phase boundary on the Nishimori line. As explained in Section 4.3, we expect it to overlap the exact $SG$ phase transition.

Contrary to what one would expect to see on the exact phase diagram [45, 46], the Nishimori line $T = T^*$ does not to cross a triple point on the RS phase diagram. The issue is that the RS phase diagram is not exact outside of the Nishimori line. In particular, the $SG$ phase boundary is not exact. Outside of the retrieval regime, the free entropy of the inverse model is the same as the direct model. Since the transition towards $gR$ of the inverse model on the Nishimori line overlaps the exact $P$-$SG$ transition of the direct model (see Section 4.3), we deduce that it must also overlap the exact $P$-$SG$ transition of the *inverse* model outside of the $gR$ phase. Plotting it on the RS phase diagrams, we see that it indeed crosses the Nishimori line and the $gR$ phase boundary at the same point, which therefore becomes a triple point, as expected.

## 4.5 Interaction order and noise tolerance

So far, we assumed that the student is informed about the interaction order used by the teacher, i.e. $p = p^*$. In this Section, we investigate the role of the student's choice of $p$ when the task is to learn from a dataset sampled by a 2-body Hopfield network, i.e. $p^* = 2$. We study two different non trivial scalings regimes of $M$ and $\beta^*$ that make pattern inference possible (see Appendix D).

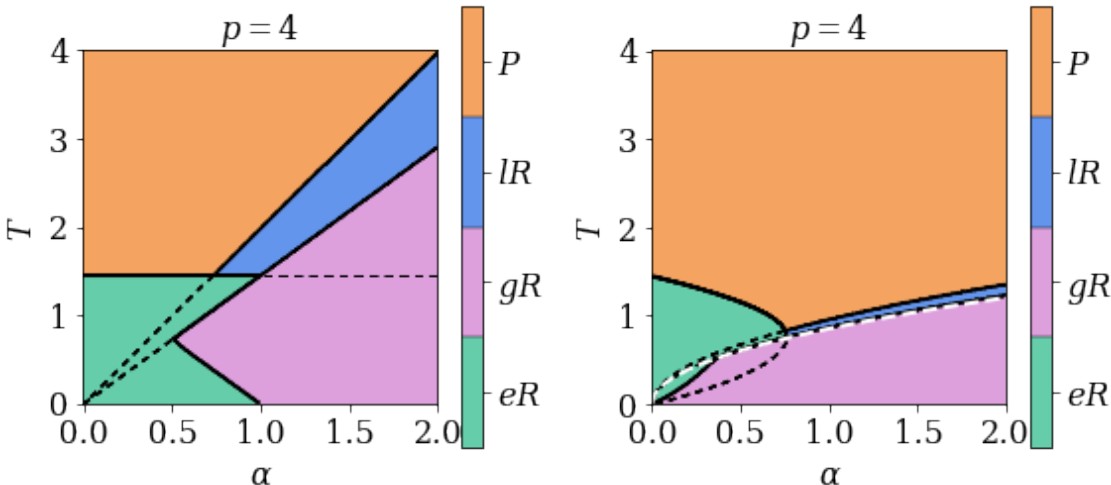

Figure 6: RS phase diagrams of inverse models with $p^* = 2$ and $p = 4$. The left plot is for $\alpha = \frac{M(p/2+1)!}{N^{p/2}}$, and $\beta^* = 1 - \frac{1}{\sqrt{2}}$ such that $\eta = 1$ and the right plot is for $\alpha = \frac{Mp!}{N^{p-1}}$ and $\beta^* = \sqrt{\frac{2\lambda}{N}}$ with $\lambda = \beta$. Accurate pattern retrieval is not possible in the paramagnetic phase ($P$) or in the example retrieval phase ($eR$), but it is possible in the local retrieval phase ($lR$) and in the global retrieval phase ($gR$). The ferromagnetic fixed point corresponding to accurate pattern retrieval is globally stable in the $gR$ phase, but locally stable in the $lR$ phase. The black dashed lines mark the metastable continuation of the $eR$, $lR$ and $gR$ phase boundaries through neighboring phases with a larger free entropy. The paramagnetic total entropy becomes negative below the white dashed line drawn on the right plot. However, the paramagnetic phase is no longer globally stable at that temperature.

### 4.5.1 Large noise scaling

We first consider a large noise scaling where $\beta^* \sim \mathcal{O}(N^{2/p-1})$ and $M \sim \mathcal{O}(N^{p-1})$, such that

$$\alpha = \frac{Mp!}{N^{p-1}}, \quad \text{and} \quad \lambda = \frac{[\beta^*]^{p/2}}{(p/2)!} N^{p/2-1},$$

are finite. In this scaling, a $p \geq 3$ network requires $\mathcal{O}(N^{p-2})$ more training examples than a $p = 2$ network with finite load $\gamma = \frac{M}{N}$, but also has a higher tolerance to teacher noise. For instance, a student with $p = 4$ interactions is able to retrieve the pattern of a teacher with $T^* \sim \mathcal{O}(N^{1/2})$ noise when it is shown enough examples $M \sim \mathcal{O}(N^3)$ to be in the $gR$ phase (see Fig. 6).

$\mathcal{O}(N^{1/2})$ noise tolerance was also observed in the $p = 4$ direct model, where it is a consequence of the redundancy stemming from storing $\mathcal{O}(N)$ memories rather than the $\mathcal{O}(N^3)$ needed to saturate the storage capacity [64]. Our $p = 4$ inverse model exploits a different kind of redundancy by learning from $\mathcal{O}(N^3)$ examples whereas $p = 2$ only needs $\mathcal{O}(N)$. In other terms, both storing extensively less memories than the maximum allowed amount and generating extensively more examples than the minimum required amount provide enough redundancy to recover a pattern muddled in an extensive amount of noise. In both cases, there is an $\mathcal{O}(N^2)$ gap between the number of patterns used in the noise-tolerant and noise-susceptible regimes. Going beyond $p = 4$, the inverse model has $\mathcal{O}(N^{1-2/p})$ noise tolerance as a function of $p$. In particular, our theory predicts that the tolerance saturates at $T^* \sim \mathcal{O}(N)$ as $p \to \infty$, but at the cost of using an intractable number of examples. This behavior is different from

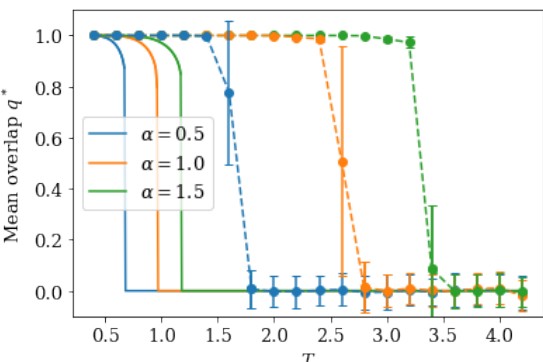

Figure 7: Monte-Carlo simulations (dashed lines) and RS saddle-point solutions (full lines) of the inverse model in the large-noise scaling with $p^* = 2$ and $p = 4$. The teacher generates $M = \frac{\alpha N^{p-1}}{p!}$ examples $\sigma^a$ with $N = 256$ components each, and the simulation results are then averaged over $L = 100$ student patterns. The student patterns are all initialized to $\xi^*$.

the $\mathcal{O}\left(N^{1/2-p/4}\right)$ tolerance of the direct $p$-body model in the noisy-learning regime studied in [65]. In other terms, the dataset noise that we are facing is of a different nature than the learning noise of [65]. In any case, it is interesting that both the direct and inverse models are able to tolerate an extensive amount of noise. Overall, our results suggest that it could be advantageous to use a student network with a relatively large $p$ to learn from a large but noisy dataset when the $p^*$ of the teacher generative model is unknown.

An unavoidable drawback of large teacher noise is that it always lead to uncorrelated examples, which makes accurate example retrieval impossible. Instead, it is replaced by the inaccurate example retrieval phase where the student has finite overlap $m$ with a noisy example generated by the teacher but no overlap with the signal (see Fig. 6). Depending on $T$ and $\alpha$, this phase can be either globally stable or locally stable. For the sake of clarity, we plot only the globally stable phase on our phase diagram in Fig. (6). The locally stable phase is arguably less important to plot because it is identical to the locally stable ferromagnetic phase previously reported in the direct model when assuming replica symmetry (see [33] and Fig. 1).

Given $m = 0$, the free entropy of the inverse model with $p \geq 3$, $p^* = 2$ and $\beta = \lambda$ is the same as on the Nishimori line (see Eq. 5 and Appendix D). As a direct consequence, the total entropy is positive outside of the *eR* phase (see Fig. 6). Additionally, the $p^* = 2$, $p \geq 3$ phase diagrams with $\beta \neq \lambda$ are identical to the $p = p^*$ phase diagrams with $\beta \neq \beta^*$, which suggests that $\beta = \lambda$ is optimal for $p^* = 2$, $p \geq 3$ in the same sense as $\beta = \beta^*$ is optimal for $p = p^*$ (see Fig. 5). Monte-Carlo simulations confirm that a student with $p \geq 3$ is able to retrieve the pattern of a teacher with $p = 2$ and $T^* \sim \mathcal{O}\left(N^{1/2}\right)$ (see Fig. 7). However, the *lR* phase transition is at a higher $T$ in the simulations than on the $\beta = \lambda$ RS phase diagram (see Fig. 5), which means that RSB is necessary to describe it accurately. One could check where replica symmetry holds by evaluating the stability of the RS saddle point throughout the phase diagram.

### 4.5.2 Finite noise scaling

We also consider a different scaling regime where $\beta^* \sim \mathcal{O}(1)$ and $M \sim \mathcal{O}\left(N^{p/2}\right)$, such that

$$\alpha = \frac{M(p/2+1)!}{N^{p/2}},$$

is finite. In this finite-noise scaling, $p \geq 3$ requires $\mathcal{O}\left(N^{p/2-1}\right)$ more training examples than $p = 2$, which is a lot less than the first scaling. For instance, a student with $p = 4$ needs $\mathcal{O}\left(N^2\right)$

examples to retrieve $\xi^*$. As before, the phase transitions are all first order, the overlap $q^*$ stays high throughout the $gR$ and $lR$ phase of $p = 4$ and $gR$ is a hard phase. The saddle-point equations (see Eqs. 6) are free from the pattern interference term $\sqrt{\alpha r}x$ present in their $p^* = p$ counterparts (see Eqs. 5) until $\beta^*$ becomes so small that is approaches $\mathcal{O}\left(N^{2/p-1}\right)$. Therefore, contrary to $p^* = p = 2$, the network is never in the $SG$ phase. Practically, it means that $p \geq 3$ gives more freedom than $p = 2$ for tuning $\beta$ and $\alpha$. The only remaining restriction is that choosing $\alpha$ and $T$ too small puts the network into the inaccurate $eR$ phase resulting from the $kz$ term (see Fig. 6). The saddle point equations can be derived without the RS ansatz because they do not involve $q$ and $r$. Consequently, we expect them to yield an exact solution. Like on the Nishimori line, the total entropy of the paramagnetic phase is always positive, which is consistent with the solution being exact.

## 4.6 Robustness against adversarial attacks

Inverse models with $p^* = 2$ and $p \geq 3$ offer an opportunity to study adversarial attacks in a simple setting because their phase diagrams have regions where the signal retrieval phases ($gR$ and $lR$) overlap with the inaccurate $eR$ phase. Recall that, in the $lR$ phase, a noisy student pattern $\xi$ either converges to $\xi^*$ or falls in the paramagnetic phase, depending on the amount of noise that $\xi$ contains initially. The quantity of noise needed to prevent pattern retrieval becomes smaller as one approaches the $lR$ to $P$ phase transition and the basin of attraction of $lR$ shrinks. Similarly, in the region of inaccurate $eR$ where signal retrieval is metastable, patterns $\xi$ that are corrupted by replacing some of their entries $\xi_i$ by the components $\sigma_i^a$ of an example $\sigma^a$ may converge to $\sigma^a$ when enough entries are replaced. The fraction $\varepsilon$ of entries that need to be replaced becomes smaller as the basin of attraction of inaccurate $eR$ expands and overtakes that of signal retrieval. In practice, an adversary can use this strategy to trick the student into converging to a pattern other than $\xi^*$. This scenario is similar to an adversarial attack targeting the input of K & H's dense Hopfield network model because the student pattern $\xi$ plays a similar role in the inverse model as the test data in K & H's dense Hopfield networks (see Fig. 3, Section 4.3 and Appendix A). In that analogy, the examples $\sigma$ are acting like the neural network weights rather than taking the role of the training data.

We will now investigate what values of the perturbation size $\varepsilon$ are a threat by deriving a formula for the largest $\varepsilon$ such that the student converges to the signal at zero temperature. This largest $\varepsilon$ will be denoted $\varepsilon^*$, and we expect it to be a good measure of adversarial robustness. The saddle-point equations with $T = 0$ indicate that the student converges to one of the signal retrieval phases if and only if $k < \eta \alpha r^*$ (see Eqs. 6). Sampling the initial conditions of $\xi_i$ from $(1 - \varepsilon)\delta\left(\xi_i - \xi_i^*\right) + \varepsilon\,\delta\left(\xi_i - \sigma_i^a\right)$ with $\varepsilon \in [0, 1]$, we get

$$r^* = p\left[\frac{1}{N}\sum_{i=1}^{(1-\varepsilon)N}\xi_i^*\xi_i^* + \frac{1}{N}\sum_{i=1}^{\varepsilon N}\xi_i^*\sigma_i^a\right]^{p-1},$$

$$k = p\left[\frac{1}{N}\sum_{i=1}^{(1-\varepsilon)N}\xi_i^*\sigma_i^a + \frac{1}{N}\sum_{i=1}^{\varepsilon N}\sigma_i^a\sigma_i^a\right]^{p-1}.$$

By the law of large numbers, $\frac{1}{\varepsilon N}\sum_{i=1}^{\varepsilon N}\xi_i^*\sigma_i^a$ and $\frac{1}{(1-\varepsilon)N}\sum_{i=1}^{(1-\varepsilon)N}\xi_i^*\sigma_i^a$ are both typically close to $m^* = \frac{1}{N}\sum_i^N \xi_i^*\sigma_i^a \approx 0$ as $N \to \infty$. If we take $\sigma^a$ to be a typical example, then $r^*$ and $k$ reduce to

$$r^* \approx p\left(1 - \varepsilon\right)^{p-1},$$

$$k \approx p\varepsilon^{p-1}.$$

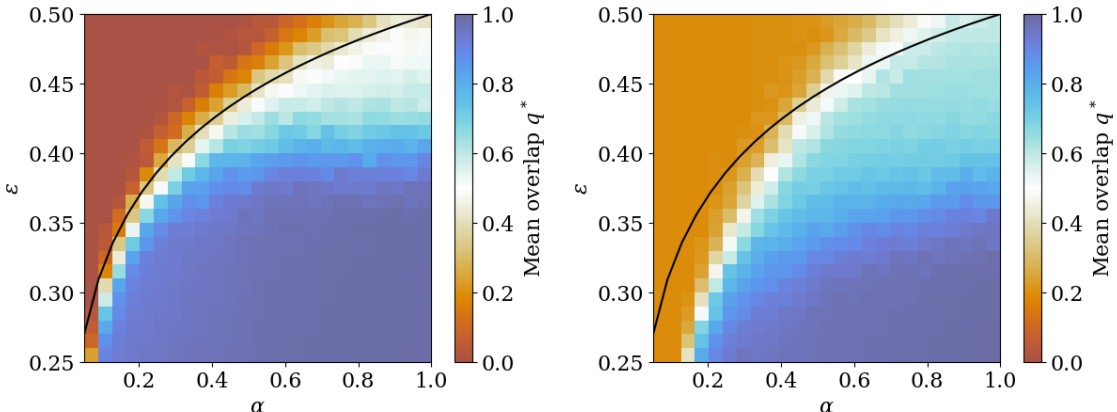

Figure 8: Monte-Carlo simulations of the overlap $q^*$ as a function of $\alpha$ and adversarial attack size $\varepsilon$ in the inverse model with $p^* = 2$, $\beta^* = 1 - \frac{1}{\sqrt{2}}$, $p = 4$, $\beta = \infty$ and $N = 1024$. The simulation results are averaged over $L = 100$ student patterns. On the left plot, the inverse model is corrupted by an example $\sigma^a$ that has a small overlap with $\xi^*$ in absolute value. On the right plot, it is corrupted by the example that has the largest overlap with $\xi^*$. The black line $\varepsilon^* = \frac{\alpha^{1/3}}{\alpha^{1/3}+1}$ is our analytical formula for the largest adversarial perturbation $\varepsilon$ such that the student retrieves $\xi^*$ rather than the example $\sigma^a$.

Substituting these expressions back in $k < \eta\alpha r^*$ yields

$$\varepsilon^{p-1} < \eta\alpha(1-\varepsilon)^{p-1} \, ,$$

$$\varepsilon < \frac{[\eta\alpha]^{\frac{1}{p-1}}}{[\eta\alpha]^{\frac{1}{p-1}} + 1} \, .$$

In other terms, the inverse model with $p^* = 2$ and even $p \geq 3$ is resistant to adversarial attacks of size $\varepsilon^* = \frac{[\eta\alpha]^{\frac{1}{p-1}}}{[\eta\alpha]^{\frac{1}{p-1}}+1}$ and smaller. For $p = 4$, $\varepsilon^*$ is in good agreement with Monte-Carlo simulations of the inverse model corrupted by a typical example (see Fig. 8). This comparison is good evidence that our solution of the finite-noise scaling is indeed exact. Additionally, $\varepsilon^*$ is a decent approximation of empirical robustness even when the inverse model is corrupted by the example that has the largest overlap with $\xi^*$. A similar construction with the perturbation sampled uniformly at random gives $k \sim \mathcal{O}\left(N^{1/2-p/2}\right) \approx 0$, so adversarial attacks are much more efficient at fooling the model than random noise. Just like adversarial attacks targeting more complicated neural networks [40, 41], our example-based attack can be hard to detect at low $\varepsilon$ because a few adversarially perturbed entries $\xi_i$ do not look very different from a low amount of meaningless noise. Moreover, $\varepsilon^*$ grows monotonically with $\alpha$, which is consistent with the common observation that larger neural networks are also more adversarially robust [43, 66–71]. At first glance, this effect can be counter-intuitive because adversarial vulnerability looks like a form of overfitting [42]. In our model, however, all examples work together to stabilize the *lR* phase, and the best way to push the student into the *eR* phase is to perturb it with a single example. Therefore, it is not surprising that increasing $\alpha$ makes the student more robust. We recall that the examples $\sigma$ are a feature-based representation of $\xi^*$. Interestingly, it means that the underlying mechanism of our example-based attack is conceptually similar to gradient-based attacks targeting many common types of neural networks [42]. In fact, gradient-based attacks find features stored in neural network weights and add them to the data in order to fool the network [42, 72–74]. It would be interesting to investigate, both empirically and theoretically, if

only a small number of weights are involved in constructing these adversarial attacks. If it is the case, it could explain why larger neural networks are often more robust. In general, we expect this kind of one-example attack to be possible in any region of signal retrieval that overlaps with the inaccurate $eR$ phase. Using $p \neq p^*$ may not be a necessary ingredient of adversarial vulnerability in more general models with other sources of mismatch, but in our case it ensures that the signal retrieval phases intersect the inaccurate $eR$ phase. Conversely, the accurate $eR$ phase is by definition robust to adversarial attacks since retrieving an example $\sigma^a$ is the same as recovering $\xi^*$. This distinction clarifies why the dense Hopfield networks designed by K & H are adversarially robust in the prototype phase despite being adversarially vulnerable in the feature phase. In fact, K & H observed that adversarial attacks are unsuccessful in the prototype phase specifically because they retrieve stored examples that are semantically meaningful [37]. In summary, our model yields two main results concerning adversarial examples. First of all, it suggests a reason why large feature-based neural networks are more adversarially robust than smaller ones. Second of all, it clarifies why dense Hopfield networks are much more robust in the prototype phase than in the feature phase.

## 5 Conclusion

In this work, we derive the exact phase diagram of the $p$-dense networks in the teacher-student setting [16, 17, 30, 38]. On the Nishimori line, we find an example retrieval phase ($eR$) and a global retrieval phase ($gR$) reminiscent of the prototype and feature regimes observed empirically in dense Hopfield networks [26]. We show that the phase transition towards $gR$ of the inverse model overlaps the paramagnetic to spin-glass ($P$-$SG$) transition of the direct model, which allows us to locate the $P$-$SG$ transition much more precisely than before [30, 33]. On the other hand, we discover that inverse models outside of the Nishimori line are able to resist an extensive amount of noise. In fact, a student with $p \geq 3$ is able to learn from a teacher with $p^* = 2$ even when the teacher's inverse temperature $\beta^*$ is as low as $\mathcal{O}\left(N^{2/p-1}\right)$. Moreover, such a student is immune to pattern interference until $\beta^*$ reaches $\mathcal{O}\left(N^{2/p-1}\right)$. In this setting, we derive a formula measuring the adversarial robustness of the student with $p \geq 3$ and $T = 0$. We then use this formula to describe how making a neural network larger can potentially increase its robustness to adversarial attacks constructed with only a few learned weights [43, 66–71]. Our model also clarifies why the prototype phase of dense Hopfield networks is adversarially robust [37]. We compare our key results against Monte-Carlo simulations.

Dense networks with exponential interactions have been argued to be the $p \to \infty$ limit of the $p$-body models [75]. It would be interesting to see if they can achieve $\mathcal{O}(N)$ noise tolerance at the cost of an exponential number of training examples. More generally, studying exponential models in the teacher-student setting would be an interesting extension of this work and could be used to complement existing studies of the direct model [75, 76]. A caveat of our model is that the teacher has only one pattern. In fact, we would need to use a teacher with at least two patterns to describe more completely the kind of adversarial attack aiming to misclassify data. It should be possible to study this kind of teacher by using an approach similar to [77]. In particular, [16] and [77] argue that the performance of restricted Boltzmann machines with a finite number $P$ of i.i.d. planted patterns is independent of $P$ in the teacher-student setting. It would be interesting to investigate whether this characteristic also holds for $p$-body dense networks. On the practical side, we highlight the untapped benefits of using $p$-body models to either resist an extensive amount of noise in the feature phase or improve adversarial robustness in the prototype phase. Overall, we stress that further investigations of dense Hopfield networks could unlock their true potential.

## Acknowledgements

**Funding information**   This work was partially supported by project SERICS (PE00000014) under the MUR National Recovery and Resilience Plan funded by the European Union - NextGenerationEU. The work was also supported by the project PRIN22TANTARI "Statistical Mechanics of Learning Machines: from algorithmic and information-theoretical limits to new biologically inspired paradigms" 20229T9EAT – CUP J53D23003640001. DT also acknowledges GNFM-Indam.

**Code availability**   The figures can be reproduced using the code available on this public Github repository.

## A  Gardner's Hamiltonian vs K & H's Hamiltonian

Consider the generalized Hopfield Hamiltonian $H[\sigma|\xi] = -\sum_{i_1 < ... < i_p = 1}^{N} J_{i_1...i_p} \sigma_{i_1}...\sigma_{i_p}$ with $p$-body interactions $J_{i_1...i_p} = \frac{p!}{N^{p-1}} \sum_{\mu=1}^{M} \xi_{i_1}^{\mu}...\xi_{i_p}^{\mu}$ described by Gardner [30], where $M$ indicates the number of patterns $\xi^{\mu}$ used to construct $J$, and $N$ denotes the number of components of each pattern $\xi^{\mu}$ and example $\sigma$. In this Section, we will omit $\xi$ in the argument of $H[\sigma|\xi]$ and write $H[\sigma]$ instead for notational simplicity. Unless indicated otherwise, we will assume a large number number of components $N \gg 1$ and patterns $M \sim \mathcal{O}(N^{p-1})$. We will start by comparing it to the dense Hopfield network Hamiltonian $\mathcal{H}[\sigma] = -\frac{1}{N^{p-1}} \sum_{\mu} (\sum_i \xi_i^{\mu} \sigma_i)^p$ studied by K & H [26].

For that purpose, we rewrite $H$ in the form $H[\sigma] = -\frac{1}{p!} \sum_{i_1 \neq ... \neq i_p} J_{i_1...i_p} \sigma_{i_1}...\sigma_{i_p}$ by summing over all permutations of $\{i_1...i_p\}$ in place of the restricted set $i_1 < ... < i_p$ and compensating for double counting with the prefactor $\frac{1}{p!}$. This manipulation leads to

$$H[\sigma] = -\frac{1}{p!} \sum_{i_1 \neq ... \neq i_p} J_{i_1...i_p} \sigma_{i_1}...\sigma_{i_p}$$
$$= -\frac{1}{N^{p-1}} \sum_{\mu} \sum_{i_1 \neq ... \neq i_p} \xi_{i_1}^{\mu}...\xi_{i_p}^{\mu} \sigma_{i_1}...\sigma_{i_p} .$$

On the other hand, K & H's Hamiltonian may be rewritten

$$\mathcal{H}[\sigma] = -\frac{1}{N^{p-1}} \sum_{\mu} \left( \sum_i \xi_i^{\mu} \sigma_i \right)^p$$
$$= -\frac{1}{N^{p-1}} \sum_{\mu} \left( \sum_{i_1} \xi_{i_1}^{\mu} \sigma_{i_1} \right)...\left( \sum_{i_p} \xi_{i_p}^{\mu} \sigma_{i_p} \right)$$
$$= -\frac{1}{N^{p-1}} \sum_{\mu} \sum_{i_1...i_p} \xi_{i_1}^{\mu}...\xi_{i_p}^{\mu} \sigma_{i_1}...\sigma_{i_p} ,$$

where the sum over $i_1...i_p$ includes both the set of indices $i_1 \neq ... \neq i_p$ found in $H[\sigma]$ and other configurations where some indices are equal. For example, the configuration $i_1 \neq ... \neq i_{p-1} = i_p$ contains the fewest equal indices after $i_1 \neq ... \neq i_p$. In other words, $\mathcal{H}[\sigma]$ can be expressed as an expansion around $H[\sigma]$, and the two Hamiltonians are equivalent when the normalized residuals $\frac{\mathcal{H}[\sigma]-H[\sigma]}{N}$ vanish in the limit of large $N$. In this study, we encounter two cases which bring different results.

**1** The Hamiltonians $\mathcal{H}[\sigma]$ and $H[\sigma]$ are dominated by a few closely packed configurations $\xi^\mu$ that have finite overlap $\frac{1}{N}\sum_i \xi_i^\mu \sigma_i \sim \mathcal{O}(1)$ with $\sigma$. We say that they are aligned with $\sigma$.

**2** The Hamiltonians $\mathcal{H}[\sigma]$ and $H[\sigma]$ are dominated by many spread out configurations $\xi^\mu$ that have microscopic overlap $\frac{1}{N}\sum_i \xi_i^\mu \sigma_i \sim \mathcal{O}(N^{-1/2})$ with $\sigma$. We say that they are misaligned with $\sigma$

We use the expansion of $\mathcal{H}[\sigma]$ to discuss both the aligned case and the misaligned case. We start by writing the $i_1 \neq ... \neq i_p$ and $i_1 \neq ... \neq i_{p-1} = i_p$ terms explicitly, which leads to the form

$$\mathcal{H}[\sigma] = -\frac{1}{N^{p-1}}\sum_\mu \sum_{i_1 \neq ... \neq i_p} \xi_{i_1}^\mu ... \xi_{i_p}^\mu \sigma_{i_1}...\sigma_{i_p}$$
$$-\frac{1}{2}\frac{p(p-1)}{N^{p-1}}\sum_\mu \sum_{i_1 \neq ... \neq i_{p-1}} \xi_{i_1}^\mu ...\left(\xi_{i_{p-1}}^\mu\right)^2 \sigma_{i_1}...\left(\sigma_{i_{p-1}}\right)^2 + ...,$$

because there are $\binom{p}{2} = \frac{p(p-1)}{2}$ ways for the indices $i_{p-1}$ and $i_p$ to be equal. This expression can be summarized by $\mathcal{H}[\sigma] = H[\sigma] + H'[\sigma] + ...$, where $H'[\sigma]$ simplifies to

$$H'[\sigma] = -\frac{1}{2}\frac{p(p-1)}{N^{p-1}}\sum_\mu \sum_{i_1 \neq ... \neq i_{p-1}} \xi_{i_1}^\mu ...\left(\xi_{i_{p-1}}^\mu\right)^2 \sigma_{i_1}...\left(\sigma_{i_{p-1}}\right)^2$$
$$= -\frac{1}{2}\frac{p(p-1)}{N^{p-2}}\sum_\mu \sum_{i_1 \neq ... \neq i_{p-2}} \xi_{i_1}^\mu ...\xi_{i_{p-2}}^\mu \sigma_{i_1}...\sigma_{i_{p-2}}$$
$$= -\frac{1}{2}\frac{p!}{N^{p-2}}\sum_\mu \sum_{i_1 < ... < i_{p-2}} \xi_{i_1}^\mu ...\xi_{i_{p-2}}^\mu \sigma_{i_1}...\sigma_{i_{p-2}}.$$

In the aligned case, $H'[\sigma]$ is $\mathcal{O}(1)$ in $N$ because the sum over $i_1 < ... < i_{p-2}$ is $\mathcal{O}(N^{p-2})$. The terms implied by the ellipsis are even smaller because their sums are resctricted by more equality constraints. Therefore, the residuals $\frac{\mathcal{H}[\sigma] - H[\sigma]}{N}$ vanish in the limit of large $N$, and the two Hamiltonians are equivalent. Conversely, we find that $\mathcal{H}[\sigma]$ and $H[\sigma]$ differ from each other in the misaligned case (see Appendix B for more details). Therefore, although the phases of $H[\sigma]$ that we obtain in this study are qualitatively similar to the ones observed by K & H [26, 37], the phase diagram of $H[\sigma]$ must be compared against a simulation of $H[\sigma]$ rather than $\mathcal{H}[\sigma]$ in order to test our theory quantitatively.

To understand how to sample $\sigma$ in both models, consider a Monte-Carlo simulation used to find the statistical equilibrium of a spin ensemble $\sigma$ with Hamiltonian $G[\sigma]$. To be more specific, suppose $\sigma$ is updated to a new state $\sigma'$ with a randomly selected spin $\sigma_i$ flipped with acceptance probability $P_i = \frac{1}{1+\exp[\beta(G[\sigma']-G[\sigma])]}$ for a large number of time-steps. This approach works well for $G[\sigma] = \mathcal{H}[\sigma]$. However, in the case of $H[\sigma]$, we find that the simulation only converges when we use the local field $h_i = \frac{p!}{N^{p-1}}\sum_\mu \xi_i^\mu \sum_{i_1 < ... < i_{p-1}} \xi_{i_1}^\mu ... \xi_{i_{p-1}}^\mu \sigma_{i_1}...\sigma_{i_{p-1}}$ mentioned by Gardner [30] to approximate $\frac{H[\sigma']-H[\sigma]}{2\sigma_i}$ at large $N$. In other words, we iteratively flip randomly chosen spins $\sigma_i$ with acceptance probability $P_i = \frac{1}{1+\exp(2\beta h_i \sigma_i)}$ for a large number of time steps. For arbitrary $p$, it is not obvious how to compute $h_i$ quickly as a sub-routine of the Monte-Carlo simulation. However, we find that both $p = 3$ and $p = 4$ have closed-formed expressions that are easy to evaluate numerically in an efficient way. To be more precise,

- $p = 3$ leads to $h_i = 3\sum_\mu \xi_i^\mu\left[\left(\frac{1}{N}\sum_j \xi_j^\mu \sigma_j\right)^2 - \frac{1}{N}\right]$,

- and $p = 4$ leads to $h_i = 4\sum_\mu \xi_i^\mu\left(\frac{1}{N}\sum_j \xi_j^\mu \sigma_j\right)\left[\left(\frac{1}{N}\sum_j \xi_j^\mu \sigma_j\right)^2 - \frac{3}{N}\right]$.

For this reason and also because the number $M \sim \mathcal{O}\left(N^{p-1}\right)$ of patterns $\xi^\mu$ used in a Monte-Carlo simulations increases exponentially with $p$, we choose to simulate only $p = 3$ and $p = 4$.

The output of the neural network model that K & H designed for classification of data is $c_j = \tanh\left[\frac{1}{2}\beta\left(\mathcal{H}\left[\sigma'\right] - \mathcal{H}\left[\sigma\right]\right)\right] \approx \tanh\left[\beta p \sum_\mu \xi_j^\mu \left(\frac{1}{N} \sum_i \xi_i^\mu \sigma_i\right)^{p-1}\right]$. We omit the linear rectifier present in the original paper [26] because the overlaps $\frac{1}{N} \sum_i \xi_i^\mu \sigma_i$ are almost always positive (see for example the Supplement of [78]). The predicted class is then $j' = \operatorname{argmax}_j \{c_j\}$. Using $1 - P_j = \frac{1}{1 + \exp[\beta(\mathcal{H}[\sigma'] - \mathcal{H}[\sigma])]}$ instead of $c_j$ does not change $j'$ because $1 - P_j$ and $c_j$ are related by $1 - P_j = \frac{1}{2}\left[c_j + 1\right]$. When we evaluate $P_i$ using $H$ instead of $\mathcal{H}$, this relation does not always hold exactly. Rather, it should be considered an approximation.

## B   Direct model cumulant expansions

In the direct model, the average replicated partition function $\left\langle Z^L \right\rangle$ takes the form:

$$\left\langle Z^L \right\rangle = \left\langle \sum_\sigma \exp\left(-\beta \sum_{\gamma=1}^L H\left[\sigma^\gamma | \xi\right]\right)\right\rangle,$$

with $\sigma = \left\{\sigma^1 \ \ldots \ \sigma^L\right\}$. Gardner simplifies it to

$$\left\langle Z^L \right\rangle \approx \left\langle \sum_\sigma \exp\left(\beta N \sum_\gamma \sum_{\mu \in \Gamma_\gamma}\left[\frac{1}{N} \sum_i \xi_i^\mu \sigma_i^\gamma\right]^p + \beta \sum_\gamma \sum_{\mu \in \bar{\Gamma}} \frac{p!}{N^{p-1}} \sum_{i_1 < \ldots < i_p} \xi_{i_1}^\mu \ldots \xi_{i_p}^\mu \sigma_{i_1}^\gamma \ldots \sigma_{i_p}^\gamma\right)\right\rangle,$$

(10)

where the sets $\Gamma_\gamma$ contain the patterns $\xi^\mu$ that have macroscopic overlap with $\sigma_\gamma$, and their complement $\bar{\Gamma} = \cap_\gamma \bar{\Gamma}_\gamma$ consists of the remaining patterns. Two approximations are used to obtain this expression:

- $\sum_{\mu \in \Gamma_\gamma} \frac{p!}{N^{p-1}} \sum_{i_1 < \ldots < i_p} \xi_{i_1}^\mu \ldots \xi_{i_p}^\mu \sigma_{i_1}^\gamma \ldots \sigma_{i_p}^\gamma \approx N \sum_{\mu \in \Gamma_\gamma}\left[\frac{1}{N} \sum_i \xi_i^\mu \sigma_i^\gamma\right]^p$ because this part of $H\left[\sigma^\gamma | \xi\right]$ is aligned with $\sigma$ (see Case **1** of Appendix A).

- $\sum_{\mu \in \bar{\Gamma}_\gamma} \frac{p!}{N^{p-1}} \sum_{i_1 < \ldots < i_p} \xi_{i_1}^\mu \ldots \xi_{i_p}^\mu \sigma_{i_1}^\gamma \ldots \sigma_{i_p}^\gamma \approx \sum_{\mu \in \bar{\Gamma}} \frac{p!}{N^{p-1}} \sum_{i_1 < \ldots < i_p} \xi_{i_1}^\mu \ldots \xi_{i_p}^\mu \sigma_{i_1}^\gamma \ldots \sigma_{i_p}^\gamma$ since $\bar{\Gamma}$ contains almost all of the elements in each $\bar{\Gamma}_\gamma$ when $N$ is large.

Gardner evaluates the contribution of the $\mu \in \bar{\Gamma}$ terms via a cumulant expansion, resulting in:

$$\log\left\langle \exp\left(\beta \sum_\gamma \frac{p!}{N^{p-1}} \sum_{i_1 < \ldots < i_p} \xi_{i_1}^\mu \ldots \xi_{i_p}^\mu \sigma_{i_1}^\gamma \ldots \sigma_{i_p}^\gamma\right)\right\rangle$$

$$\approx \beta \left\langle \sum_\gamma \frac{p!}{N^{p-1}} \sum_{i_1 < \ldots < i_p} \xi_{i_1}^\mu \ldots \xi_{i_p}^\mu \sigma_{i_1}^\gamma \ldots \sigma_{i_p}^\gamma\right\rangle + \frac{1}{2}\beta^2 \left\langle \left[\sum_\gamma \frac{p!}{N^{p-1}} \sum_{i_1 < \ldots < i_p} \xi_{i_1}^\mu \ldots \xi_{i_p}^\mu \sigma_{i_1}^\gamma \ldots \sigma_{i_p}^\gamma\right]^2\right\rangle$$

$$\approx \frac{1}{2}\beta^2 \left\langle \left[\sum_\gamma \frac{p!}{N^{p-1}} \sum_{i_1 < \ldots < i_p} \xi_{i_1}^\mu \ldots \xi_{i_p}^\mu \sigma_{i_1}^\gamma \ldots \sigma_{i_p}^\gamma\right]\left[\sum_\delta \frac{p!}{N^{p-1}} \sum_{j_1 < \ldots < j_p} \xi_{j_1}^\mu \ldots \xi_{j_p}^\mu \sigma_{j_1}^\delta \ldots \sigma_{j_p}^\delta\right]\right\rangle,$$

because the product of independent spins $\xi^\mu_{i_1}...\xi^\mu_{i_p}$ averages to 0. The sums are then regrouped to get

$$\log\left\langle \exp\left(\beta \sum_\gamma \frac{p!}{N^{p-1}} \sum_{i_1<...<i_p} \xi^\mu_{i_1}...\xi^\mu_{i_p}\sigma^\gamma_{i_1}...\sigma^\gamma_{i_p}\right)\right\rangle$$

$$= \frac{1}{2}\beta^2\left[\frac{p!}{N^{p-1}}\right]^2\left\langle \sum_\gamma\sum_\delta \sum_{i_1<...<i_p}\sum_{j_1<...<j_p} \xi^\mu_{i_1}\xi^\mu_{j_1}...\xi^\mu_{i_p}\xi^\mu_{j_p}\; \sigma^\gamma_{i_1}\sigma^\delta_{j_1}...\sigma^\gamma_{i_p}\sigma^\delta_{j_p}\right\rangle .$$

Consider $\xi^\mu_i \xi^\mu_j$ for an arbitrary pair of indices $i$ and $j$. There are two cases.

- If $i = j$, then $\xi^\mu_i \xi^\mu_j$ is deterministic and equal to 1.

- If $i \neq j$, then $\xi^\mu_i \xi^\mu_j$ can be either $+1$ and $-1$ with equal probabilities.

On the one hand, if $i_n = j_n$ for all $n$, then $\left\langle \xi^\mu_{i_1}\xi^\mu_{j_1}...\xi^\mu_{i_p}\xi^\mu_{j_p}\right\rangle = 1$. On the other hand, if $i_n \neq j_n$ for some $n$, then $\left\langle \xi^\mu_{i_1}\xi^\mu_{j_1}...\xi^\mu_{i_p}\xi^\mu_{j_p}\right\rangle = 0$ because $\xi^\mu_{i_1}\xi^\mu_{j_1}...\xi^\mu_{i_p}\xi^\mu_{j_p}$ is still a product of independent random spins once the deterministic variables are removed. These two cases can be summarized by $\left\langle \xi^\mu_{i_1}\xi^\mu_{j_1}...\xi^\mu_{i_p}\xi^\mu_{j_p}\right\rangle = \delta_{i_1 j_1}...\delta_{i_p j_p}$, which then gives

$$\log\left\langle \exp\left(\beta \sum_\gamma \frac{p!}{N^{p-1}} \sum_{i_1<...<i_p} \xi^\mu_{i_1}...\xi^\mu_{i_p}\sigma^\gamma_{i_1}...\sigma^\gamma_{i_p}\right)\right\rangle$$

$$= \frac{1}{2}\beta^2\left[\frac{p!}{N^{p-1}}\right]^2\sum_\gamma\sum_\delta \sum_{i_1<...<i_p}\sum_{j_1<...<j_p} \delta_{i_1 j_1}...\delta_{i_p j_p}\; \sigma^\gamma_{i_1}\sigma^\delta_{j_1}...\sigma^\gamma_{i_p}\sigma^\delta_{j_p}$$

$$= \frac{1}{2}\beta^2\left[\frac{p!}{N^{p-1}}\right]^2\sum_\gamma\sum_\delta \sum_{i_1<...<i_p} \sigma^\gamma_{i_1}\sigma^\delta_{i_1}...\sigma^\gamma_{i_p}\sigma^\delta_{i_p}$$

$$\approx \frac{1}{2}\beta^2\frac{p!}{N^{p-1}}\frac{1}{N^{p-1}}\sum_{\gamma\delta}\left[\sum_i \sigma^\gamma_i\sigma^\delta_i\right]^p$$

$$= \beta^2\frac{p!}{N^{p-1}}N\sum_{\gamma<\delta}\left[\frac{1}{N}\sum_i \sigma^\gamma_i\sigma^\delta_i\right]^p + \frac{1}{2}\beta^2\frac{p!}{N^{p-1}}LN .$$

The order $n > 2$ terms are subdominant in $N$ and can be neglected when $p \geq 3$ [30]. The RS free entropy is then obtained through a standard approach to the replica method. Note that Gardner's Hamiltonian is misaligned with $\sigma$ when the free entropy is dominated by this cumulant expansion (see Case **2** of Appendix A). In the case of K & H's Hamiltonian, we must also take into account the correction $H'[\sigma] = \frac{1}{2}\frac{p!}{N^{p-2}}\sum_\gamma\sum_{i_1<...<i_{p-2}} \xi^\mu_{i_1}...\xi^\mu_{i_{p-2}}\sigma^\gamma_{i_1}...\sigma^\gamma_{i_{p-2}}$ introduced in appendix A by imposing $i_{p-1} = i_p$. In fact, a cumulant expansion of this expression gives

$$\log\left\langle \exp\left(\beta p \sum_\gamma \frac{1}{2}\frac{p!}{N^{p-2}} \sum_{i_1<...<i_{p-2}} \xi^\mu_{i_1}...\xi^\mu_{i_{p-2}}\sigma^\gamma_{i_1}...\sigma^\gamma_{i_{p-2}}\right)\right\rangle$$

$$\approx \frac{1}{4}\beta^2\frac{p!}{N^{p-2}}\frac{p(p-1)}{N^{p-2}}\sum_{\gamma<\delta}\left[\sum_i \sigma^\gamma_i\sigma^\delta_i\right]^{p-2} + \frac{1}{8}\beta^2\frac{p!}{N^{p-2}}L$$

$$= \frac{1}{4}p(p-1)\beta^2\frac{p!}{N^{p-1}}N\sum_{\gamma<\delta}\left[\frac{1}{N}\sum_i \sigma^\gamma_i\sigma^\delta_i\right]^{p-2} + \frac{1}{8}\beta^2\frac{p!}{N^{p-1}}LN ,$$

which contributes to the free energy on the same order in $N$ as Gardner's Hamiltonian. Therefore, K & H's Hamiltonian is not equivalent to Gardner's Hamiltonian when the latter is misaligned with $\sigma$ (see Case 2). The index configurations with more equality constraints also contribute to the free entropy on the same order in $N$ because the factors of $N$ that are lost to equality constraints are restored when the sums get squared in the cumulant expansion.

$p = 2$ is the only positive integer such that Gardner's Hamiltonian and Krotov's Hamiltonian are equivalent [5, 30]. In the misaligned case with a single stored pattern $\xi^*$ (see Case 2), the free entropy of $p = 2$ simplifies to

$$
\begin{aligned}
\frac{\log(Z)}{N} &= \frac{1}{N} \log \left\langle \exp \left\{ \beta \frac{2}{N} \sum_{i_1 < i_2} \xi_{i_1}^* \xi_{i_2}^* \sigma_{i_1}^\gamma \sigma_{i_2}^\gamma \right\} \right\rangle + \log 2 \\
&= \frac{1}{N} \log \left\langle \exp(-\beta) \int_\mathbb{R} dx \frac{1}{\sqrt{2\pi}} \exp \left\{ -\frac{1}{2} x^2 + x \sqrt{\beta \frac{2}{N}} \sum_i \xi_i^* \sigma_i^\gamma \right\} \right\rangle + \log 2 \\
&= \frac{1}{N} \log \left[ \int_\mathbb{R} dx \frac{1}{\sqrt{2\pi}} \exp \left\{ -\frac{1}{2} x^2 \right\} \cosh^N \left( x \sqrt{\beta \frac{2}{N}} \right) \right] - \beta \frac{1}{N} + \log 2,
\end{aligned}
$$

by using the Hubbard-Stratonovich transformation. At large $N$, it approximates to:

$$
\begin{aligned}
\frac{\log(Z)}{N} &\approx \frac{1}{N} \log \left[ \int dx \frac{1}{\sqrt{2\pi}} \exp \left\{ -\frac{1}{2} x^2 \right\} \left( 1 + \beta \frac{1}{N} x^2 \right)^N \right] - \beta \frac{1}{N} + \log 2 \\
&\approx \frac{1}{N} \log \left[ \int_\mathbb{R} dx \frac{1}{\sqrt{2\pi}} \exp \left\{ -\frac{1}{2} x^2 \right\} \exp(\beta x^2) \right] - \beta \frac{1}{N} + \log 2 \\
&= \left( -\frac{1}{2} \log(1 - 2\beta) - \beta \right) \frac{1}{N} + \log 2,
\end{aligned}
$$

thanks to the well-known limit $\lim_{N \to \infty} \left( 1 + \frac{1}{N} z \right)^N = \exp(z)$. This free entropy is consistent with the one found in literature when $\alpha = \frac{1}{N}$ [5].

## C  Teacher-student replicated partition function

Recall that the student samples its pattern from the posterior $P(\xi | \sigma) = \frac{P(\xi) \prod_a P(\sigma^a | \xi)}{P(\sigma)}$ (see Section 3). Given $P(\xi)$ uniform, it can be rewritten as $P(\xi | \sigma) = \frac{\prod_a P(\sigma^a | \xi)}{\sum_\xi \prod_a P(\sigma^a | \xi)}$, where $P(\sigma^a | \xi)$ is the distribution of the direct model with a single pattern $\xi$. To simplify $P(\xi | \sigma)$ further, we need to manipulate the partition function $Z = \sum_{\sigma^a} \exp(-\beta H[\sigma^a | \xi])$ of $P(\sigma^a | \xi)$ (see Appendix A for the definition of $H[\sigma | \xi]$). Under the gauge transformation $\sigma_i^a \to \xi_i \sigma_i^a$, we may write

$$
Z = \sum_{\sigma^a} \exp \left( \beta \frac{p!}{N^{p-1}} \sum_{i_1 < ... < i_p} \sigma_{i_1}^a ... \sigma_{i_p}^a \right),
$$

without changing the configurations of $\sigma^a$ that we are summing over. Therefore, $Z$ does not depend on $\xi$, and we can factor it out of the sum $\sum_\xi$, which yields

$$
P(\xi|\sigma) = \frac{\prod_a \frac{1}{Z} \exp\left(\beta \frac{p!}{N^{p-1}} \sum_{i_1 < \ldots < i_p} \xi_{i_1} \ldots \xi_{i_p} \sigma^a_{i_1} \ldots \sigma^a_{i_p}\right)}{\sum_\xi \prod_a \frac{1}{Z} \exp\left(\beta \frac{p!}{N^{p-1}} \sum_{i_1 < \ldots < i_p} \xi_{i_1} \ldots \xi_{i_p} \sigma^a_{i_1} \ldots \sigma^a_{i_p}\right)}
$$

$$
= \frac{\exp\left(\beta \frac{p!}{N^{p-1}} \sum_a \sum_{i_1 < \ldots < i_p} \xi_{i_1} \ldots \xi_{i_p} \sigma^a_{i_1} \ldots \sigma^a_{i_p}\right)}{\sum_\xi \exp\left(\beta \frac{p!}{N^{p-1}} \sum_a \sum_{i_1 < \ldots < i_p} \xi_{i_1} \ldots \xi_{i_p} \sigma^a_{i_1} \ldots \sigma^a_{i_p}\right)}.
$$

Therefore, we define the partition function of the inverse model to be $\mathcal{Z} = \sum_\xi \exp(-\beta H[\xi|\sigma])$ (again, see Appendix A for the definition of $H[\xi|\sigma]$). The $L^{\text{th}}$ power of $\mathcal{Z}$ and its average then take the form

$$
\mathcal{Z}^L = \sum_\xi \prod_b \exp\left(\beta \frac{p!}{N^{p-1}} \sum_a \sum_{i_1 < \ldots < i_p} \xi^b_{i_1} \ldots \xi^b_{i_p} \sigma^a_{i_1} \ldots \sigma^a_{i_p}\right),
$$

$$
\langle \mathcal{Z}^L \rangle = \sum_\sigma P(\sigma) \sum_\xi \exp\left(\beta \frac{p!}{N^{p-1}} \sum_{ab} \sum_{i_1 < \ldots < i_p} \xi^b_{i_1} \ldots \xi^b_{i_p} \sigma^a_{i_1} \ldots \sigma^a_{i_p}\right),
$$

where $b \in \{1 \ldots L\}$ label replicas in the set of patterns $\xi = \{\xi^1 \ldots \xi^L\}$ inferred by the student. Using the definition of conditional probability, we rewrite $P(\sigma)$ as

$$
P(\sigma) = \sum_{\xi^*} P(\sigma|\xi^*) P(\xi^*)
$$

$$
= \frac{1}{2^N} \sum_{\xi^*} P(\sigma|\xi^*)
$$

$$
= \frac{1}{2^N} \sum_{\xi^*} \prod_a P(\sigma^a|\xi^*),
$$

where $P(\sigma|\xi^*)$ has the same functional form as $P(\sigma|\xi^b)$, but has hyperparameters $p^*$ and $\beta^*$ in place of $p$ and $\beta$. As we did for $Z$, we factor the partition function $Z^*$ of $P(\sigma^a|\xi^*)$ out of the sum, which yields

$$
P(\sigma) = \frac{1}{2^N} \frac{1}{[Z^*]^M} \sum_{\xi^*} \prod_a \exp\left(\beta^* \frac{p^*!}{N^{p^*-1}} \sum_{i_1 < \ldots < i_{p^*}} \xi^*_{i_1} \ldots \xi^*_{i_{p^*}} \sigma^a_{i_1} \ldots \sigma^a_{i_{p^*}}\right)
$$

$$
= \frac{1}{2^N} \frac{\mathcal{Z}^*}{[Z^*]^M} = \frac{1}{2^{MN}} \frac{\mathcal{Z}^*}{\left[2^{N/M-N} Z^*\right]^M},
$$

where $\mathcal{Z}^* = \sum_{\xi^*} \exp(-\beta^* H[\xi^*|\sigma])$ is the partition function of the inverse model with interaction order $p^*$. Using $\sum_\sigma P(\sigma) = 1$, we immediately deduce that $\left[2^{N/M-N} Z^*\right]^M = \langle \mathcal{Z}^* \rangle$. Plugging $P(\sigma) = \frac{1}{2^{MN}} \frac{\mathcal{Z}^*}{\langle \mathcal{Z}^* \rangle}$ back in $\langle \mathcal{Z}^L \rangle$ then gives

$$
\langle \mathcal{Z}^L \rangle = \frac{1}{2^{MN}} \frac{1}{\langle \mathcal{Z}^* \rangle} \sum_{\xi^*} \sum_\sigma \exp\left(\beta^* \frac{p^*!}{N^{p^*-1}} \sum_a \sum_{i_1 < \ldots < i_{p^*}} \xi^*_{i_1} \ldots \xi^*_{i_{p^*}} \sigma^a_{i_1} \ldots \sigma^a_{i_{p^*}}\right)
$$

$$
\sum_\xi \exp\left(\beta \frac{p!}{N^{p-1}} \sum_{ab} \sum_{i_1 < \ldots < i_p} \xi^b_{i_1} \ldots \xi^b_{i_p} \sigma^a_{i_1} \ldots \sigma^a_{i_p}\right).
$$

We simplify this expression to:

$$
\begin{aligned}
\left\langle \mathcal{Z}^L \right\rangle &= \frac{1}{2^{MN}} \frac{1}{\langle \mathcal{Z}^* \rangle} \sum_{\xi^*} \sum_{\sigma} \exp\Bigg( \beta^* \frac{p^*!}{N^{p^*-1}} \sum_{a \in \Gamma_*} \sum_{i_1 < \ldots < i_{p^*}} \xi_{i_1}^* \ldots \xi_{i_{p^*}}^* \sigma_{i_1}^a \ldots \sigma_{i_{p^*}}^a \\
&\quad + \beta^* \frac{p^*!}{N^{p^*-1}} \sum_{a \in \bar\Gamma_*} \sum_{i_1 < \ldots < i_{p^*}} \xi_{i_1}^* \ldots \xi_{i_{p^*}}^* \sigma_{i_1}^a \ldots \sigma_{i_{p^*}}^a \Bigg) \\
&\quad \sum_{\xi} \exp\Bigg( \beta \frac{p!}{N^{p-1}} \sum_b \sum_{a \in \Gamma_b} \sum_{i_1 < \ldots < i_p} \xi_{i_1}^b \ldots \xi_{i_p}^b \sigma_{i_1}^a \ldots \sigma_{i_p}^a \\
&\quad + \beta \frac{p!}{N^{p-1}} \sum_b \sum_{a \in \bar\Gamma_b} \sum_{i_1 < \ldots < i_p} \xi_{i_1}^b \ldots \xi_{i_p}^b \sigma_{i_1}^a \ldots \sigma_{i_p}^a \Bigg) \\
&\approx \frac{1}{2^{MN}} \frac{1}{\langle \mathcal{Z}^* \rangle} \sum_{\xi^*} \sum_{\sigma} \exp\Bigg( \beta^* N \sum_{a \in \Gamma_*} \left[ \frac{1}{N} \sum_i \xi_i^* \sigma_i^a \right]^{p^*} \\
&\quad + \beta^* \frac{p^*!}{N^{p^*-1}} \sum_{a \in \bar\Gamma} \sum_{i_1 < \ldots < i_{p^*}} \xi_{i_1}^* \ldots \xi_{i_{p^*}}^* \sigma_{i_1}^a \ldots \sigma_{i_{p^*}}^a \Bigg) \\
&\quad \sum_{\xi} \exp\Bigg( \beta N \sum_b \sum_{a \in \Gamma_b} \left[ \frac{1}{N} \sum_i \xi_i^b \sigma_i^a \right]^p \\
&\quad + \beta \frac{p!}{N^{p-1}} \sum_b \sum_{a \in \bar\Gamma} \sum_{i_1 < \ldots < i_p} \xi_{i_1}^b \ldots \xi_{i_p}^b \sigma_{i_1}^a \ldots \sigma_{i_p}^a \Bigg),
\end{aligned}
$$

where $\Gamma_b$ represents the set of inputs $\sigma^a$ which have macroscopic overlap with the pattern $\xi^b$, and $\bar\Gamma = \left[ \cap_b \bar\Gamma_b \right] \cap \bar\Gamma_*$ contains almost all of the elements in each $\bar\Gamma_b$ and $\bar\Gamma_*$ for $N \to \infty$. The reasoning used to build the sets $\Gamma_*$, $\Gamma_b$ and $\bar\Gamma$ is the same as outlined at the start of appendix B.

## D Teacher-student free entropy

Assuming that the teacher is misaligned with $\sigma$ (see Case 2 of Appendix A), the form of $\left\langle \mathcal{Z}^L \right\rangle$ obtained in appendix C simplifies to

$$
\begin{aligned}
\left\langle \mathcal{Z}^L \right\rangle &\approx \frac{1}{2^{MN}} \frac{1}{\langle \mathcal{Z}^* \rangle} \sum_{\xi^* \xi} \sum_{\sigma} \exp\Bigg( \beta^* \frac{p^*!}{N^{p^*-1}} \sum_{a \in \bar\Gamma} \sum_{i_1 < \ldots < i_{p^*}} \xi_{i_1}^* \ldots \xi_{i_{p^*}}^* \sigma_{i_1}^a \ldots \sigma_{i_{p^*}}^a \Bigg) \\
&\quad \exp\Bigg( \beta N \sum_b \sum_{a \in \Gamma_b} \left[ \frac{1}{N} \sum_i \xi_i^b \sigma_i^a \right]^p + \beta \frac{p!}{N^{p-1}} \sum_b \sum_{a \in \bar\Gamma} \sum_{i_1 < \ldots < i_p} \xi_{i_1}^b \ldots \xi_{i_p}^b \sigma_{i_1}^a \ldots \sigma_{i_p}^a \Bigg).
\end{aligned}
$$

In order to evaluate $\langle \mathcal{Z}^* \rangle = \left[ 2^{N/M-N} Z^* \right]^M$, we recall that the teacher is a special case of the direct model with a single memory (see Section 3). Since the teacher is in the misaligned case, its free entropy is

$$
\frac{\log(Z^*)}{N} = \begin{cases} \left( -\frac{1}{2} \log(1 - 2\beta^*) - \beta^* \right) \frac{1}{N} + \log 2, & p^* = 2, \\ \frac{1}{2} [\beta^*]^2 \frac{p^*!}{N^{p^*-1}} + \log 2 + \mathcal{O}\left( \frac{1}{N^{3p^*/2-2}} \right), & p^* \geq 3, \end{cases}
$$

as derived in Appendix B. Given $\alpha^* = \frac{M p^*!}{N^{p^*-1}}$, we use it to simplify $\frac{\log \langle \mathcal{Z}^* \rangle}{N}$ to

$$
\frac{\log \langle \mathcal{Z}^* \rangle}{N} = \frac{M \log \left[ 2^{N/M - N} Z^* \right]}{N}
$$

$$
= \begin{cases} \frac{1}{2} \left( -\frac{1}{2} \log(1 - 2\beta^*) - \beta^* \right) \alpha^* + \log 2, & p^* = 2, \\ \frac{1}{2} [\beta^*]^2 \alpha^* + \log 2 + \mathcal{O}\left( \frac{1}{N^{p^*/2-1}} \right), & p^* \geq 3, \end{cases}
$$

which is the paramagnetic free entropy of a $p^*$-body Hopfield network [5, 30]. Coming back to $\langle \mathcal{Z}^L \rangle$, we fix order parameters $q^{*b}$, $q^{bc}$ and $m_a^b$ using the delta functions $\delta \left( N q^{*b} - \sum_i \xi_i^* \xi_i^b \right)$, $\delta \left( N q^{bc} - \sum_i \xi_i^b \xi_i^c \right)$ and $\delta \left( N m_a^b - \sum_i \xi_i^b \sigma_i^a \right)$, which results in

$$
\langle \mathcal{Z}^L \rangle = \frac{1}{2^{MN}} \frac{1}{\langle \mathcal{Z}^* \rangle} \sum_{\xi^* \xi} \sum_\sigma \int_{\mathbb{R}} \prod_b dq^{*b} \prod_{b<c} dq^{bc} \prod_b \prod_{a \in \Gamma_b} dm_a^b
$$

$$
\delta \left( N q^{*b} - \sum_i \xi_i^* \xi_i^b \right) \delta \left( N q^{bc} - \sum_i \xi_i^b \xi_i^c \right) \delta \left( N m_a^b - \sum_i \xi_i^b \sigma_i^a \right)
$$

$$
\exp \Bigg( \beta N \sum_b \sum_{a \in \Gamma_b} \left[ \frac{1}{N} \sum_i \xi_i^b \sigma_i^a \right]^p
$$

$$
+ \beta^* \frac{p^*!}{N^{p^*-1}} \sum_{a \in \bar{\Gamma}} \sum_{i_1 < \dots < i_{p^*}} \xi_{i_1}^* \dots \xi_{i_{p^*}}^* \sigma_{i_1}^a \dots \sigma_{i_{p^*}}^a
$$

$$
+ \beta \frac{p!}{N^{p-1}} \sum_b \sum_{a \in \bar{\Gamma}} \sum_{i_1 < \dots < i_p} \xi_{i_1}^b \dots \xi_{i_p}^b \sigma_{i_1}^a \dots \sigma_{i_p}^a \Bigg).
$$

In Fourier space, this expression takes the form

$$
\langle \mathcal{Z}^L \rangle = \frac{1}{\langle \mathcal{Z}^* \rangle} \sum_{\xi^* \xi} \Bigg\langle \int \prod_b dq^{*b} dr^{*b} \prod_{b<c} dq^{bc} dr^{bc} \prod_b \prod_{a \in \Gamma_b} dm_a^b dk_a^b
$$

$$
\exp \left\{ \beta^* \beta \alpha \sum_b \left( \sum_i \xi_i^* \xi_i^b - N q^{*b} \right) r^{*b} + \beta^2 \alpha \sum_{b<c} \left( \sum_i \xi_i^b \xi_i^c - N q^{bc} \right) r^{bc} \right\}
$$

$$
\exp \left\{ \beta \sum_b \sum_{a \in \Gamma_b} \left( \sum_i \xi_i^b \sigma_i^a - N m_a^b \right) k_a^b + \beta N \sum_b \sum_{a \in \Gamma_b} \left[ \frac{1}{N} \sum_i \xi_i^b \sigma_i^a \right]^p \right.
$$

$$
+ \beta^* \frac{p^*!}{N^{p^*-1}} \sum_{a \in \bar{\Gamma}} \sum_{i_1 < \dots < i_{p^*}} \xi_{i_1}^* \dots \xi_{i_{p^*}}^* \sigma_{i_1}^a \dots \sigma_{i_{p^*}}^a
$$

$$
\left. + \beta \frac{p!}{N^{p-1}} \sum_b \sum_{a \in \bar{\Gamma}} \sum_{i_1 < \dots < i_p} \xi_{i_1}^b \dots \xi_{i_p}^b \sigma_{i_1}^a \dots \sigma_{i_p}^a \right\} \Bigg\rangle_\sigma,
$$

where the sum over $\sigma$ with a pre-factor of $\frac{1}{2^{MN}}$ was replaced by the uniform average $\langle \rangle_\sigma$. Following the same reasoning as in appendix B, a second order cumulant expansion of the last

two terms for any $a \in \bar{\Gamma}$ yields

$$\log \left\langle \exp \left\{ \beta^* \frac{p^*!}{N^{p^*-1}} \sum_{i_1 < ... < i_{p^*}} \xi^*_{i_1} ... \xi^*_{i_{p^*}} \sigma^a_{i_1} ... \sigma^a_{i_{p^*}} + \beta \frac{p!}{N^{p-1}} \sum_b \sum_{i_1 < ... < i_p} \xi^b_{i_1} ... \xi^b_{i_p} \sigma^a_{i_1} ... \sigma^a_{i_p} \right\} \right\rangle$$

$$\approx \frac{1}{2} \beta^2 \left[ \frac{p!}{N^{p-1}} \right]^2 \sum_{b \neq c} \sum_{i_1 < ... < i_p} \sum_{j_1 < ... < j_p} \xi^b_{i_1} \xi^c_{j_1} ... \xi^b_{i_p} \xi^c_{j_p} \left\langle \sigma^a_{i_1} \sigma^a_{j_1} ... \sigma^a_{i_p} \sigma^a_{j_p} \right\rangle$$

$$+ \beta^* \beta \frac{p^*!}{N^{p^*-1}} \frac{p!}{N^{p-1}} \sum_b \sum_{i_1 < ... < i_{p^*}} \sum_{j_1 < ... < j_p} \left\langle \xi^*_{i_1} \sigma^a_{i_1} ... \xi^*_{i_{p^*}} \sigma^a_{i_{p^*}} \xi^b_{j_1} \sigma^a_{j_1} ... \xi^b_{j_p} \sigma^a_{j_p} \right\rangle$$

$$+ \frac{1}{2} \beta^2 \frac{p!}{N^{p-1}} L N + \frac{1}{2} [\beta^*]^2 \frac{p^*!}{N^{p^*-1}} N .$$

When $p^* = p$, it reduces to

$$\log \left\langle \exp \left\{ \beta^* \frac{p^*!}{N^{p^*-1}} \sum_{i_1 < ... < i_{p^*}} \xi^*_{i_1} ... \xi^*_{i_{p^*}} \sigma^a_{i_1} ... \sigma^a_{i_{p^*}} + \beta \frac{p!}{N^{p-1}} \sum_b \sum_{i_1 < ... < i_p} \xi^b_{i_1} ... \xi^b_{i_p} \sigma^a_{i_1} ... \sigma^a_{i_p} \right\} \right\rangle$$

$$= \beta^2 \frac{p!}{N^{p-1}} N \sum_{b < c} \left[ \frac{1}{N} \sum_i \xi^b_i \xi^c_i \right]^p + \beta^* \beta \frac{p!}{N^{p-1}} N \sum_b \left[ \frac{1}{N} \sum_i \xi^*_i \xi^b_i \right]^p$$

$$+ \frac{1}{2} \beta^2 \frac{p!}{N^{p-1}} L N + \frac{1}{2} [\beta^*]^2 \frac{p!}{N^{p-1}} N ,$$

because $\left\langle \sigma^a_{i_n} \sigma^a_{j_n} \right\rangle = \delta_{i_n j_n}$ (see Appendix B for more details). On the contrary, the second order expectation $\left\langle \xi^*_{i_1} \sigma^a_{i_1} ... \xi^*_{i_{p^*}} \sigma^a_{i_{p^*}} \xi^b_{j_1} \sigma^a_{j_1} ... \xi^b_{j_p} \sigma^a_{j_p} \right\rangle$ vanishes when $p^* \neq p$. In fact, spins come in pairs $\left\langle \sigma^a_{i_n} \sigma^a_{j_n} \right\rangle = \delta_{i_n j_n}$ only up to $n \leq \min \{p^*, p\}$, and the remaining single-spin averages $\left\langle \sigma^a_{i_n} \right\rangle = 0$ make the second order expectation vanish.

We need to go beyond second order to treat $p^* \neq p$. We will focus on $p^* = 2$ and $p \geq 3$ to investigate the consequences of using a *p*-body model to learn examples generated by the original 2-body Hopfield model. For simplicity, we take $p$ even so that the spins of both terms can be grouped in pairs at order $\frac{p}{2} + 1$, when the teacher term $\beta^* \frac{2}{N} \sum_{i_1 < i_2} \xi^*_{i_1} \xi^*_{i_2} \sigma^a_{i_1} \sigma^a_{i_2}$ is raised to the power of $\frac{p}{2}$ and the student term is raised to the power of 1. This restriction will simplify some of the incoming calculations. To leading order in $N$, the cumulant generating function reduces to

$$\log \left\langle \exp \left\{ \beta^* \frac{2}{N} \sum_{i_1 < i_2} \xi^*_{i_1} \xi^*_{i_2} \sigma^a_{i_1} \sigma^a_{i_2} + \beta \frac{p!}{N^{p-1}} \sum_b \sum_{i_1 < ... < i_p} \xi^b_{i_1} ... \xi^b_{i_p} \sigma^a_{i_1} ... \sigma^a_{i_p} \right\} \right\rangle$$

$$\approx \log \left[ \left\langle \exp \left\{ \beta^* \frac{2}{N} \sum_{i_1 < i_2} \xi^*_{i_1} \xi^*_{i_2} \sigma^a_{i_1} \sigma^a_{i_2} \right\} \right\rangle \right.$$

$$\left\langle \exp \left\{ \beta \frac{p!}{N^{p-1}} \sum_b \sum_{i_1 < ... < i_p} \xi^b_{i_1} ... \xi^b_{i_p} \sigma^a_{i_1} ... \sigma^a_{i_p} \right\} \right\rangle$$

$$+ \left\langle \beta \frac{p!}{N^{p-1}} \sum_b \sum_{j_1 < ... < j_p} \xi^b_{j_1} ... \xi^b_{j_p} \sigma^a_{j_1} ... \sigma^a_{j_p} \exp \left\{ \beta^* \frac{2}{N} \sum_{i_1 < i_2} \xi^*_{i_1} \xi^*_{i_2} \sigma^a_{i_1} \sigma^a_{i_2} \right\} \right\rangle \right] ,$$

where the last term encompasses the teacher-student coupling that allows retrieval to take place. The teacher term

$$\log \left\langle \exp \left\{ \beta^* \frac{2}{N} \sum_{i_1 < i_2} \xi^*_{i_1} \xi^*_{i_2} \sigma^a_{i_1} \sigma^a_{i_2} \right\} \right\rangle \approx -\frac{1}{2} \log (1 - 2\beta^*) - \beta^* ,$$

and the student term

$$\log\left\langle \exp\left\{\beta\frac{p!}{N^{p-1}}\sum_b\sum_{i_1<...<i_p}\xi_{i_1}^b...\xi_{i_p}^b\sigma_{i_1}^a...\sigma_{i_p}^a\right\}\right\rangle$$

$$\approx \beta^2\frac{p!}{N^{p-1}}N\sum_{b<c}\left[\frac{1}{N}\sum_i\xi_i^b\xi_i^c\right]^p+\frac{1}{2}\beta^2\frac{p!}{N^{p-1}}LN\,,$$

are both known from Appendix B. Later on, we will use $\log(z^*)$ and $z^*$ as shorthands for $-\frac{1}{2}\log(1-2\beta^*)-\beta^*$ and $\exp\left(-\frac{1}{2}\log(1-2\beta^*)-\beta^*\right)$, respectively. The coupling between the teacher and the student can be rewritten as

$$\left\langle \beta\frac{p!}{N^{p-1}}\sum_b\sum_{j_1<...<j_p}\xi_{j_1}^b...\xi_{j_p}^b\sigma_{j_1}^a...\sigma_{j_p}^a\exp\left\{\beta^*\frac{2}{N}\sum_{i_1<i_2}\xi_{i_1}^*\xi_{i_2}^*\sigma_{i_1}^a\sigma_{i_2}^a\right\}\right\rangle$$

$$=\left\langle \beta\frac{p!}{N^{p-1}}\sum_b\sum_{j_1<...<j_p}\xi_{j_1}^*...\xi_{j_p}^*\xi_{j_1}^b...\xi_{j_p}^b\,\xi_{j_1}^*...\xi_{j_p}^*\sigma_{j_1}^a...\sigma_{j_p}^a\exp\left\{\beta^*\frac{2}{N}\sum_{i_1<i_2}\xi_{i_1}^*\xi_{i_2}^*\sigma_{i_1}^a\sigma_{i_2}^a\right\}\right\rangle$$

$$=\beta\frac{p!}{N^{p-1}}\sum_b\sum_{j_1<...<j_p}\xi_{j_1}^*...\xi_{j_p}^*\xi_{j_1}^b...\xi_{j_p}^b\left\langle \xi_{j_1}^*...\xi_{j_p}^*\sigma_{j_1}^a...\sigma_{j_p}^a\exp\left\{\beta^*\frac{2}{N}\sum_{i_1<i_2}\xi_{i_1}^*\xi_{i_2}^*\sigma_{i_1}^a\sigma_{i_2}^a\right\}\right\rangle\,,$$

because $\left[\xi_{j_n}^*\right]^2=1$ for every index $j_n$. All interacting spin tuples of the form $\xi_{j_1}^*...\xi_{j_p}^*\sigma_{j_1}^a...\sigma_{i_p}^a$ are statistically equivalent as long as $j_1<...<j_p$, so the teacher-student coupling simplifies to

$$\left\langle \beta\frac{p!}{N^{p-1}}\sum_b\sum_{j_1<...<j_p}\xi_{j_1}^b...\xi_{j_p}^b\sigma_{j_1}^a...\sigma_{j_p}^a\exp\left\{\beta^*\frac{2}{N}\sum_{i_1<i_2}\xi_{i_1}^*\xi_{i_2}^*\sigma_{i_1}^a\sigma_{i_2}^a\right\}\right\rangle$$

$$=\beta\frac{p!}{N^{p-1}}\sum_b\sum_{i_1<...<i_p}\xi_{i_1}^*...\xi_{i_p}^*\xi_{i_1}^b...\xi_{i_p}^b$$

$$\left\langle \frac{p!}{N^p}\sum_{j_1<...<j_p}\xi_{j_1}^*...\xi_{j_p}^*\sigma_{j_1}^a...\sigma_{i_p}^a\exp\left\{\beta^*\frac{2}{N}\sum_{i_1<i_2}\xi_{i_1}^*\xi_{i_2}^*\sigma_{i_1}^a\sigma_{i_2}^a\right\}\right\rangle$$

$$=V(\beta^*,p)\beta\frac{p!}{N^{p-1}}\sum_b\sum_{i_1<...<i_p}\xi_{i_1}^*...\xi_{i_p}^*\xi_{i_1}^b...\xi_{i_p}^b\,,$$

where $V(\beta^*,p)=\left\langle \frac{p!}{N^p}\sum_{j_1<...<j_p}\xi_{j_1}^*...\xi_{j_p}^*\sigma_{j_1}^a...\sigma_{i_p}^a\exp\left(\beta^*\frac{2}{N}\sum_{i_1<i_2}\xi_{i_1}^*\xi_{i_2}^*\sigma_{i_1}^a\sigma_{i_2}^a\right)\right\rangle$ does not depend on the microscopic details of the system. In fact, it can be expressed as a combination of the moments of $z^*$, which can all be derived from $\log(z^*)$. To leading order in $N$, the cumulant generating function expands to

$$\log\left\langle \exp\left\{\beta^*\frac{2}{N}\sum_{i_1<i_2}\xi_{i_1}^*\xi_{i_2}^*\sigma_{i_1}^a\sigma_{i_2}^a+\beta\frac{p!}{N^{p-1}}\sum_b\sum_{i_1<...<i_p}\xi_{i_1}^b...\xi_{i_p}^b\sigma_{i_1}^a...\sigma_{i_p}^a\right\}\right\rangle$$

$$\approx -\frac{1}{2}\log(1-2\beta^*)-\beta^*+\beta^2\frac{p!}{N^{p-1}}N\sum_{b<c}\left[\frac{1}{N}\sum_i\xi_i^b\xi_i^c\right]^p+\frac{1}{2}\beta^2\frac{p!}{N^{p-1}}LN$$

$$+[1-2\beta^*]^{1/2}\exp(\beta^*)V(\beta^*,p)\beta N\sum_b\left[\frac{1}{N}\sum_i\xi_i^*\xi_i^b\right]^p\,.$$

At this stage, we only need to find $V(\beta^*,p)$ in order to solve the system. We focus on two different scalings of $M$ and $\beta^*$ that make the teacher-student coupling leading order in $N$:

**1** $M \sim \mathcal{O}\left(N^{p-1}\right)$ and $\beta^* \sim \mathcal{O}\left(N^{2/p-1}\right)$ will be called the large-noise scaling.

**2** $M \sim \mathcal{O}\left(N^{p/2}\right)$ and $\beta^* \sim \mathcal{O}(1)$ will be called the finite-noise scaling.

The student term vanishes in the first scenario but is leading order in the second one. The case of the teacher-student coupling is more subtle. When $\beta^*$ is small, we may keep only the first non-vanishing order of the exponential function present in the definition of $V(\beta^*, p)$. Since $p$ is even, it leads to

$$
\begin{aligned}
V(\beta^*, p) &\approx \frac{1}{(p/2)!} \left\langle \frac{p!}{N^p} \sum_{j_1 < \ldots < j_p} \xi^*_{j_1} \ldots \xi^*_{j_p} \sigma^a_{j_1} \ldots \sigma^a_{j_p} \left( \beta^* \frac{2}{N} \sum_{i_1 < i_2} \xi^*_{i_1} \xi^*_{i_2} \sigma^a_{i_1} \sigma^a_{i_2} \right)^{p/2} \right\rangle \\
&= \frac{[\beta^*]^{p/2}}{(p/2)!} \frac{2^{p/2}}{N^{p/2}} \frac{p!}{2^{p/2}} \\
&= \frac{[\beta^*]^{p/2}}{(p/2)!} \frac{p!}{N^{p/2}},
\end{aligned}
\tag{11}
$$

because there are $\prod_{n=1}^{p/2} \binom{2n}{2} = \frac{p!}{2^{p/2}}$ spin pairings with non-zero expectation that satisfy the inequality constraints. In the large-noise scaling, we set

$$
\lambda = \frac{[\beta^*]^{p/2}}{(p/2)!} N^{p/2-1} \sim \mathcal{O}(1),
$$

to get the asymptotically exact expression $V\left([(p/2)!]^{2/p} N^{1-2/p}, p\right) = \lambda \frac{p!}{N^{p-1}}$. In the finite-noise scaling, this expansion is only an order of magnitude approximation. However, it still indicates that $V(\beta^*, p)$ is $\mathcal{O}\left(N^{-p/2}\right)$ when $\beta^*$ is $\mathcal{O}(1)$ in $N$. In other words, it shows that there is an $\mathcal{O}(1)$ parameter $\eta$ such that $V(\beta^*(\eta, p), p) = \eta \frac{(p/2+1)!}{N^{p/2}}$. We will now use the cumulants $\frac{\partial \log(z^*)}{\partial \beta^*}$ and $\frac{\partial \log(z^*)}{\partial \beta^{*2}}$ of $z^*$ to derive the value of $\eta$ corresponding to $p = 4$. First of all, note that $\frac{4!}{N^4} \sum_{j_1 < \ldots < j_4} \xi^*_{j_1} \ldots \xi^*_{j_4} \sigma^a_{j_1} \ldots \sigma^a_{j_4}$ can be expressed as:

$$
\begin{aligned}
&\frac{24}{N^4} \sum_{j_1 < j_2 < j_3 < j_4} \xi^*_{j_1} \xi^*_{j_2} \xi^*_{j_3} \xi^*_{j_4} \sigma^a_{j_1} \sigma^a_{j_2} \sigma^a_{j_3} \sigma^a_{j_4} \\
&= \frac{1}{N^4} \sum_{j_1 \neq j_2 \neq j_3 \neq j_4} \xi^*_{j_1} \xi^*_{j_2} \xi^*_{j_3} \xi^*_{j_4} \sigma^a_{j_1} \sigma^a_{j_2} \sigma^a_{j_3} \sigma^a_{j_4} \\
&= \frac{1}{N^4} \left[ \sum_{j_1 \neq j_2} \xi^*_{j_1} \xi^*_{j_2} \sigma^a_{j_1} \sigma^a_{j_2} \right] \left[ \sum_{j_3 \neq j_4} \xi^*_{j_3} \xi^*_{j_4} \sigma^a_{j_3} \sigma^a_{j_4} \right] - \frac{4}{N^3} \left[ \sum_{j_1 \neq j_2} \xi^*_{j_1} \xi^*_{j_2} \sigma^a_{j_1} \sigma^a_{j_2} \right] - \frac{2}{N^2} \\
&= \frac{1}{N^2} \left[ \frac{2}{N} \sum_{j_1 < j_2} \xi^*_{j_1} \xi^*_{j_2} \sigma^a_{j_1} \sigma^a_{j_2} \right]^2 - \frac{4}{N^2} \left[ \frac{2}{N} \sum_{j_1 < j_2} \xi^*_{j_1} \xi^*_{j_2} \sigma^a_{j_1} \sigma^a_{j_2} \right] - \frac{2}{N^2},
\end{aligned}
$$

by subtracting the diagonals where pairs of indices are equal. Therefore, $\frac{1}{z^*} V(\beta^*, p)$ reduces to

$$
\frac{1}{z^*} V(\beta^*, p) = \left\langle \frac{24}{N^4} \sum_{j_1 < j_2 < j_3 < j_4} \xi^*_{j_1} \xi^*_{j_2} \xi^*_{j_3} \xi^*_{j_4} \sigma^a_{i_1} \sigma^a_{i_2} \sigma^a_{i_3} \sigma^a_{i_4} \exp\left\{ \beta^* \frac{2}{N} \sum_{i_1 < i_2} \xi^*_{i_1} \xi^*_{i_2} \sigma^a_{i_1} \sigma^a_{i_2} \right\} \right\rangle
$$

$$
= \frac{1}{z^*} \frac{1}{N^2} \left[ \left\langle \left[ \frac{2}{N} \sum_{j_1 < j_2} \xi^*_{j_1} \xi^*_{j_2} \sigma^a_{i_1} \sigma^a_{i_2} \right]^2 \exp\left\{ \beta^* \frac{2}{N} \sum_{i_1 < i_2} \xi^*_{i_1} \xi^*_{i_2} \sigma^a_{i_1} \sigma^a_{i_2} \right\} \right\rangle \right.
$$

$$
-4 \left\langle \left[ \frac{2}{N} \sum_{j_1 < j_2} \xi^*_{j_1} \xi^*_{j_2} \sigma^a_{i_1} \sigma^a_{i_2} \right] \exp\left\{ \beta^* \frac{2}{N} \sum_{i_1 < i_2} \xi^*_{i_1} \xi^*_{i_2} \sigma^a_{i_1} \sigma^a_{i_2} \right\} \right\rangle
$$

$$
-2 \left\langle \exp\left\{ \beta^* \frac{2}{N} \sum_{i_1 < i_2} \xi^*_{i_1} \xi^*_{i_2} \sigma^a_{i_1} \sigma^a_{i_2} \right\} \right\rangle \right]
$$

$$
= \frac{1}{N^2} \left[ \frac{\partial \log(z^*)}{\partial \beta^{*2}} + \left[ \frac{\partial \log(z^*)}{\partial \beta^*} \right]^2 - 4 \frac{\partial \log(z^*)}{\partial \beta^*} - 2 \right].
$$

The cumulants evaluate to

$$
\frac{\partial \log(z^*)}{\partial \beta^*} = \frac{\partial}{\partial \beta^*} \left[ -\frac{1}{2} \log(1 - 2\beta^*) - \beta^* \right] = \frac{2\beta^*}{1 - 2\beta^*},
$$

$$
\frac{\partial \log(z^*)}{\partial \beta^{*2}} = \frac{\partial}{\partial \beta^{*2}} \left[ -\frac{1}{2} \log(1 - 2\beta^*) - \beta^* \right] = \frac{2}{(1 - 2\beta^*)^2},
$$

so we obtain

$$
\frac{1}{z^*} V(\beta^*, p) = \frac{1}{N^2} \left[ \frac{2}{(1 - 2\beta^*)^2} + \frac{4[\beta^*]^2}{(1 - 2\beta^*)^2} - \frac{8\beta^*}{1 - 2\beta^*} - 2 \right]
$$

$$
= \frac{6}{N^2} \frac{2[\beta^*]^2}{(1 - 2\beta^*)^2}.
$$

In other terms, we find $\eta = \frac{2[\beta^*]^2}{(1 - 2\beta^*)^2}$ when $p = 4$. In summary, depending on the scaling, the teacher student coupling either simplifies to

**1** $\beta \lambda \alpha \frac{N}{M} \sum_b \left[ \frac{1}{N} \sum_i \xi^*_i \xi^b_i \right]^p$ where $\alpha = \frac{Mp!}{N^{p-1}}$ and $\lambda = \frac{[\beta^*]^{p/2}}{(p/2)!} N^{p/2 - 1}$ are finite,

**2** or $\beta \eta \alpha \frac{N}{M} \sum_b \left[ \frac{1}{N} \sum_i \xi^*_i \xi^b_i \right]^p$ where $\alpha = \frac{M(p/2 + 1)!}{N^{p/2}}$ and $\eta$ are finite.

In either case, the result is similar to $p^* = p$ except for its pre-factor. We describe the rest of the derivation only for $p^* = p$ because the $p^* = 2$ and $p \geq 3$ calculations are almost identical. Putting the result of the $p^* = p$ cumulant expansion back in $\langle \mathcal{Z}^L \rangle$, we get:

$$
\langle \mathcal{Z}^L \rangle \approx \frac{1}{\langle \mathcal{Z}^* \rangle} \sum_{\xi^* \xi} \left\langle \int \prod_b dq^{*b} dr^{*b} \prod_{b<c} dq^{bc} dr^{bc} \prod_b \prod_{a \in \Gamma_b} dm^b_a \, dk^b_a \right.
$$

$$
\exp\left\{ \beta^* \beta \alpha \sum_b \left( \sum_i \xi^*_i \xi^b_i - N q^{*b} \right) r^{*b} + \beta^2 \alpha \sum_{b<c} \left( \sum_i \xi^b_i \xi^c_i - N q^{bc} \right) r^{bc} \right\}
$$

$$
\exp\left\{ \beta \sum_b \sum_{a \in \Gamma_b} \left( \sum_i \xi^b_i \sigma^a_i - N m^b_a \right) k^b_a + \beta N \sum_b \sum_{a \in \Gamma_b} [m^b_a]^p \right\}
$$

$$
\exp\left\{ \beta^* \beta \alpha N \sum_b [q^{*b}]^p + \beta^2 \alpha N \sum_{b<c} [q^{bc}]^p + \frac{1}{2} \beta^2 \alpha L N + \frac{1}{2} [\beta^*]^2 \alpha N \right\} \right\rangle,
$$

where $\alpha = \frac{Mp!}{N^{p-1}}$. The saddle point of $\left\langle \mathcal{Z}^L \right\rangle$ then evaluates to

$$
\begin{aligned}
\frac{\log\left\langle \mathcal{Z}^L \right\rangle}{N} \approx \underset{m,k,q,r,q^*,r^*}{\mathrm{Extr}} \Bigg[ & \beta^* \beta \alpha \sum_b \left[ q^{*b} \right]^p + \beta^2 \alpha \sum_{b<c} \left[ q^{bc} \right]^p + \beta \sum_b \sum_{a \in \Gamma_b} \left[ m_a^b \right]^p \\
& - \beta^* \beta \alpha \sum_b r^{*b} q^{*b} - \beta^2 \alpha \sum_{b<c} r^{bc} q^{bc} - \beta \sum_b \sum_{a \in \Gamma_b} m_a^b k_a^b \\
& + \frac{1}{2} \beta^2 \alpha L + \frac{1}{2} [\beta^*]^2 \alpha - \frac{\log\langle \mathcal{Z} \rangle}{N} + \log 2 \\
& + \frac{1}{N} \log \Bigg\langle \sum_\xi \exp \Bigg\{ \beta \sum_b \sum_{a \in \Gamma_b} k_a^b \sum_i \xi_i^b \sigma_i^a \\
& + \beta^* \beta \alpha \sum_b r^{*b} \sum_i \xi_i^* \xi_i^b + \beta^2 \alpha \sum_{b<c} r^{bc} \sum_i \xi_i^b \xi_i^c \Bigg\} \Bigg\rangle_{\xi^* \sigma} \Bigg] \\
= \mathrm{Extr} \Bigg[ & \beta^* \beta \alpha \sum_b \left[ q^{*b} \right]^p + \beta^2 \alpha \sum_{b<c} \left[ q^{bc} \right]^p + \beta \sum_b \sum_{a \in \Gamma_b} \left[ m_a^b \right]^p \\
& - \beta^* \beta \alpha \sum_b r^{*b} q^{*b} - \beta^2 \alpha \sum_{b<c} r^{bc} q^{bc} - \beta \sum_b \sum_{a \in \Gamma_b} m_a^b k_a^b \\
& + \frac{1}{2} \beta^2 \alpha L + \frac{1}{2} [\beta^*]^2 \alpha - \frac{\log\langle \mathcal{Z} \rangle}{N} + \log 2 \\
& + \frac{1}{N} \sum_i \log \Bigg\langle \sum_{\xi_i} \exp \Bigg\{ \beta \sum_b \sum_{a \in \Gamma_b} k_a^b \xi_i^b \sigma_i^a \\
& + \beta^* \beta \alpha \sum_b r^{*b} \xi_i^* \xi_i^b + \beta^2 \alpha \sum_{b<c} r^{bc} \xi_i^b \xi_i^c \Bigg\} \Bigg\rangle_{\xi_i^* \sigma_i} \Bigg],
\end{aligned}
$$

where the average over $\xi^*$ and $\sigma$ is uniform. We use $\frac{\log\langle \mathcal{Z}^* \rangle}{N} = \frac{1}{2} [\beta^*]^2 \alpha + \log 2$ to simplify $\frac{\log\left\langle \mathcal{Z}^L \right\rangle}{N}$ to

$$
\begin{aligned}
\frac{\log\left\langle \mathcal{Z}^L \right\rangle}{N} \approx \mathrm{Extr} \Bigg[ & \beta^* \beta \alpha \sum_b \left[ q^{*b} \right]^p + \beta^2 \alpha \sum_{b<c} \left[ q^{bc} \right]^p + \beta \sum_b \sum_{a \in \Gamma_b} \left[ m_a^b \right]^p \\
& - \beta^* \beta \alpha \sum_b r^{*b} q^{*b} - \beta^2 \alpha \sum_{b<c} r^{bc} q^{bc} - \beta \sum_b \sum_{a \in \Gamma_b} m_a^b k_a^b \\
& + \frac{1}{2} \beta^2 \alpha L + \frac{1}{N} \sum_i \log \Bigg\langle \sum_{\xi_i} \exp \Bigg\{ \beta \sum_b \sum_{a \in \Gamma_b} k_a^b \xi_i^b \sigma_i^a \\
& + \beta^* \beta \alpha \sum_b r^{*b} \xi_i^* \xi_i^b + \beta^2 \alpha \sum_{b<c} r^{bc} \xi_i^b \xi_i^c \Bigg\} \Bigg\rangle_{\xi_i^* \sigma_i} \Bigg].
\end{aligned}
$$

Assuming each $\xi^b$ has macroscopic overlap with at most one pattern $\sigma^a$ and using the replica-symmetric ansatz $q^{*b} = q^*$, $q^{bc} = q$, $r^{*b} = r^*$, $r^{bc} = r$, $m_a^b = m$, $k_a^b = k$, the free entropy

approximates to

$$f = \lim_{N \to \infty, L \to 0} \left( \frac{\partial}{\partial L} \left[ \frac{1}{N} \log \langle Z^L \rangle \right] \right)$$

$$\approx \text{Extr} \left[ \beta^* \beta \alpha [q^*]^p - \frac{1}{2} \beta^2 \alpha q^p + \beta m^p - \beta^* \beta \alpha r^* q^* + \frac{1}{2} \beta^2 \alpha r q \right.$$

$$- \beta m k + \frac{1}{2} \beta^2 \alpha + \lim_{L \to 0} \left( \frac{\partial}{\partial L} \left[ \log \left\langle \sum_{\xi_i} \exp \left\{ \beta k \sum_b \xi_i^b \sigma_i^a \right. \right. \right. \right.$$

$$\left. \left. \left. \left. + \beta^* \beta \alpha r^* \sum_b \xi_i^* \xi_i^b + \beta^2 \alpha r \sum_{b<c} \xi_i^b \xi_i^c \right\} \right\rangle \right] \right) \right].$$

Furthermore, the Hubbard-Stratonovich transformation gives

$$\exp \left\{ \beta^2 \alpha r \sum_{b<c} \xi_i^b \xi_i^c \right\} \propto \exp \left\{ -\frac{1}{2} \beta^2 \alpha r L \right\} \int_{\mathbb{R}} dx \exp \left\{ -\frac{1}{2} x^2 + x \beta \sqrt{\alpha r} \sum_b \xi_i^b \right\}.$$

We can then use the factorization

$$\sum_{\xi_i} \exp \left\{ \beta k \sum_b \xi_i^b \sigma_i^a + \beta^* \beta \alpha r^* \sum_b \xi_i^* \xi_i^b + x \beta \sqrt{\alpha r} \sum_b \xi_i^b \right\}$$

$$= \prod_b \sum_{\xi_i^b} \exp \left\{ \beta k \xi_i^b \sigma_i^a + \beta^* \beta \alpha r^* \xi_i^* \xi_i^b + x \beta \sqrt{\alpha r} \xi_i^b \right\}$$

$$= \prod_b \left[ 2 \cosh \left( \beta k \sigma_i^a + \beta^* \beta \alpha r^* \xi_i^* + x \beta \sqrt{\alpha r} \right) \right]$$

$$= 2^L \cosh^L \left( \beta k \sigma_i^a + \beta^* \beta \alpha r^* \xi_i^* + x \beta \sqrt{\alpha r} \right),$$

in order to simplify the free energy to

$$f = \text{Extr} \left\{ \beta^* \beta \alpha [q^*]^p - \frac{1}{2} \beta^2 \alpha q^p + \beta m^p - \beta^* \beta \alpha r^* q^* + \frac{1}{2} \beta^2 \alpha r q - \beta m k \right.$$

$$- \frac{1}{2} \beta^2 \alpha r + \frac{1}{2} \beta^2 \alpha + \lim_{L \to 0} \left( \frac{\partial}{\partial L} \left[ \log \left\langle \sum_{\xi_i} \int_{\mathbb{R}} dx \exp \left\{ -\frac{1}{x} x^2 \right\} \right. \right. \right.$$

$$\left. \left. \left. \exp \left\{ \beta k \sum_b \xi_i^b \sigma_i^a + \beta^* \beta \alpha r^* \sum_b \xi_i^* \xi_i^b + x \beta \sqrt{\alpha r} \sum_b \xi_i^b \right\} \right\rangle \right] \right) \right\}$$

$$= \text{Extr} \left\{ \beta^* \beta \alpha [q^*]^p - \frac{1}{2} \beta^2 \alpha q^p + \beta m^p - \beta^* \beta \alpha r^* q^* + \frac{1}{2} \beta^2 \alpha r q - \beta m k \right.$$

$$- \frac{1}{2} \beta^2 \alpha r + \frac{1}{2} \beta^2 \alpha + \log 2 + \lim_{L \to 0} \left( \frac{\partial}{\partial L} \left[ \log \left\langle \int_{\mathbb{R}} dx \exp \left\{ -\frac{1}{x} x^2 \right\} \right. \right. \right.$$

$$\left. \left. \left. \cosh^L \left( \beta k \sigma_i^a + \beta^* \beta \alpha r^* \xi_i^* + x \beta \sqrt{\alpha r} \right) \right\rangle \right] \right) \right\}.$$

After differentiating and taking the limit, we get

$$f = \text{Extr}_{m,k,q,r,q^*,r^*} \left\{ \beta^* \beta \alpha [q^*]^p - \frac{1}{2} \beta^2 \alpha q^p + \beta m^p - \beta^* \beta \alpha r^* q^* \right.$$

$$+ \frac{1}{2} \beta^2 \alpha r q - \frac{1}{2} \beta^2 \alpha r - \beta m k + \frac{1}{2} \beta^2 \alpha + \log 2$$

$$\left. + \int dx \frac{1}{\sqrt{2\pi}} \exp \left\{ -\frac{1}{2} x^2 \right\} \left\langle \log \left[ \cosh \left( \beta \left[ \sqrt{\alpha r} x + \beta^* \alpha r^* + k z \right] \right) \right] \right\rangle_z \right\}.$$

In the case of $p^* = 2$ and $p \geq 3$ with finite $\alpha = \frac{Mp!}{N^{p-1}}$ and $\lambda = \frac{[\beta^*]^{p/2}}{(p/2)!} N^{p/2-1}$, the free energy has the same form but with $\beta^*$ replaced by $\lambda$. On the other other hand, the free energy with finite $\alpha = \frac{M(p/2+1)!}{N^{p/2}}$ and $\eta$ evaluates to:

$$f = \operatorname*{Extr}_{m,k,q^*,r^*} \left\{ \beta \eta \alpha [q^*]^p - \beta m^p - \beta \eta \alpha r^* q^* - \beta m k + \log 2 + \left\langle \log \left[ \cosh \left( \beta \left[ \eta \alpha r^* + k z \right] \right) \right] \right\rangle_z \right\}.$$

## E   Direct model RSB ansatz

Recall that the average replicated partition function of the direct model (see Eq. 10) takes the form

$$\langle Z^L \rangle \approx \left\langle \sum_\sigma \exp \left( \beta N \sum_\gamma \sum_{\mu \in \Gamma_\gamma} \left[ \frac{1}{N} \sum_i \xi_i^\mu \sigma_i^\gamma \right]^p + \beta \sum_\gamma \sum_{\mu \in \bar\Gamma} \frac{p!}{N^{p-1}} \sum_{i_1 < \dots < i_p} \xi_{i_1}^\mu \dots \xi_{i_p}^\mu \sigma_{i_1}^\gamma \dots \sigma_{i_p}^\gamma \right) \right\rangle. \quad (12)$$

Introducing a new replica $\sigma^0$, we rewrite it as

$$\langle Z^L \rangle = \left\langle \sum_\sigma \exp \left( \beta N \sum_\gamma \sum_{\mu \in \Gamma_\gamma} \left[ \frac{1}{N} \sum_i \xi_i^\mu \sigma_i^\gamma \right]^p + \beta \sum_\gamma \sum_{\mu \in \bar\Gamma} \frac{p!}{N^{p-1}} \sum_{i_1 < \dots < i_p} \xi_{i_1}^\mu \dots \xi_{i_p}^\mu \sigma_{i_1}^\gamma \dots \sigma_{i_p}^\gamma \right) \frac{\langle Z \rangle}{\langle Z \rangle} \right\rangle,$$

where $Z = \sum_{\sigma_0} \exp \left( \beta \sum_{\mu \in \bar\Gamma} \frac{p!}{N^{p-1}} \sum_{i_1 < \dots < i_p} \xi_{i_1}^\mu \dots \xi_{i_p}^\mu \sigma_{i_1}^0 \dots \sigma_{i_p}^0 \right)$. Recall that, in the paramagnetic phase, we have (see [30] and also Appendix B)

$$\langle Z \rangle = \exp \left( \frac{1}{2} \beta^2 \alpha + \log 2 + \mathcal{O} \left( \frac{1}{N^{p/2-2}} \right) \right) = Z \exp \left( \mathcal{O} \left( \frac{1}{N^{p/2-2}} \right) \right),$$

so $\langle Z^L \rangle$ can be expressed as

$$\langle Z^L \rangle = \frac{1}{\langle Z \rangle} \left\langle \sum_\sigma \exp \left( \beta N \sum_\gamma \sum_{\mu \in \Gamma_\gamma} \left[ \frac{1}{N} \sum_i \xi_i^\mu \sigma_i^\gamma \right]^p + \beta \sum_\gamma \sum_{\mu \in \bar\Gamma} \frac{p!}{N^{p-1}} \sum_{i_1 < \dots < i_p} \xi_{i_1}^\mu \dots \xi_{i_p}^\mu \sigma_{i_1}^\gamma \dots \sigma_{i_p}^\gamma \right) \right.$$
$$\left. \sum_{\sigma_0} \exp \left( \beta \sum_{\mu \in \bar\Gamma} \frac{p!}{N^{p-1}} \sum_{i_1 < \dots < i_p} \xi_{i_1}^\mu \dots \xi_{i_p}^\mu \sigma_{i_1}^0 \dots \sigma_{i_p}^0 + \mathcal{O} \left( \frac{1}{N^{p/2-2}} \right) \right) \right\rangle.$$

The $\mathcal{O} \left( \frac{1}{N^{p/2-2}} \right)$ corrections vanish to leading order in $N$ when we calculate the free entropy.

## F  Monte-Carlo simulations for various system sizes

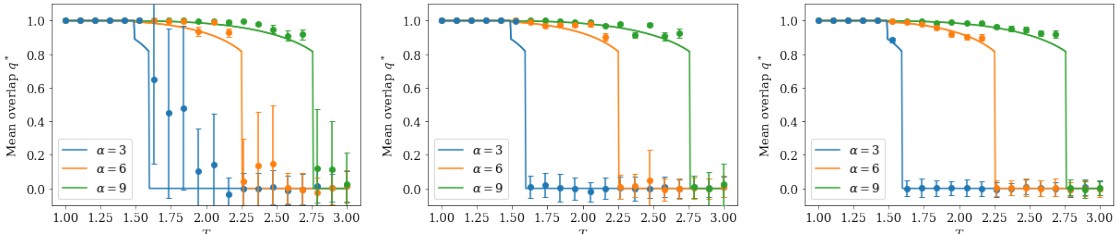

Figure 9: Monte-Carlo simulations of the $p = 3$ inverse model compared against saddle-point solutions for different values of $N$. The $lR$ phase is not included in these plots. The left plot has $N = 128$, the center plot has $N = 256$, and the right plot has $N = 512$. The dots are simulation data at a few values of $\alpha$, and the lines are slices of the saddle-point solutions at the same $\alpha$. There are $M = \frac{\alpha N^{p-1}}{p!}$ examples $\sigma^a$, and simulation results are averaged over $L = 100$ student patterns. The simulation data is sometimes systematically shifted up with respect to the saddle-point solution, but the size of the difference tends to decrease with $N$. The shift is the most visible when $\alpha = 6$ and right after the fall from $eR$ to $gR$ when $\alpha = 3$. As expected, the fluctuations of the paramagnetic phase also decrease with $N$.

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
