# Peer review of "Dense Hopfield Networks in the Teacher-Student Setting"

_SciPost Physics, doi:SciPost Phys. 17, 040 (2024)_

## Round 1 · Referee Report · Anonymous (Referee 2) · 2024-5-3

Strengths

I believe that the work represents a timely study, given the current interest of many different scientific communities on the class of models and questions addressed in the manuscript.

Weaknesses

I believe that the structure of the presentation could be developed a little bit more. The authors explore many phase diagrams and many settings corresponding to several configurations of the control parameters and it could be much better (in my opinion) to structure the manuscript in small sections highlighting the way in which control parameters are chosen and changed.

Report

In the present manuscript the authors study a particular class of Dense Hopfield Networks in the teacher-student setting which allows to study the properties of these models either as memorization models or generative models.

The particular class of networks on which the manuscript focuses on are defined by Hamiltonians involving degrees of freedom which interact via p-body interaction terms.

Previous works were focused on the standard regime where $p=2$ and the present work reports a generalization of these results to the case $p\geq 3$.

I believe that the work represents a timely study, given the current interest of many different scientific communities on the class of models and questions addressed in the manuscript.

I have some questions and remarks, listed below, which I would suggest to address in the revised version of the manuscript.

1) At the end of page 4 the authors argue that the SG transition happen when the entropy of the paramagnetic phase becomes negative. Why is this so? Similarly: in figure 1, how is the line between the P phase and SG phase computed? Can the authors detail how these phase diagrams are derived?

2) At the beginning of page 5, lines 129-134 it is mentioned that the model is 1RSB. Is this true everywhere in the phase diagram or the authors expect fullRSB at sufficiently low temperature?

3) Lines 186-193: why it is not possible to have a phase with $q^*\neq 0$ and $m\neq 0$ at the same time?

4) Given that there is a first order transition from the paramagnetic to the retrieval phase (see the discussion between lines 300 and 317), one would expect a "dynamical 1RSB phase". At the same time this would suggest that the direct model has a SG phase of 1RSB type with a dynamical 1RSB transition at higher temperatures. Could the authors comment on this point?

Requested changes

1) Reference 26 is always referred as Krotov's work despite the fact that 26 has two authors. It would be more fair to refer to 26 as "KH" work for simplicity.

2) In the caption of Fig.2 the authors could emphasize that the phase diagrams are on the Nishimori line.

3) The Theta Heaviside function in line 222 is misleading: Maybe the authors could use $\theta(q-1)$ rather than $\theta(q)$

Recommendation

Ask for minor revision

  • validity: -
  • significance: -
  • originality: -
  • clarity: -
  • formatting: -
  • grammar: -

Author:  Robin Thériault  on 2024-05-21  [id 4503]

(in reply to Report 2 on 2024-05-03)
Category:
answer to question

First of all we thank the referee for carefully reading the manuscript and for the insightful comments. Following the very first suggestion we divided the text into smaller subsection to hopefully improve readability. Concerning the list of remarks, a point-by-point answer follows:

1) We thank the referee for the comment that allows us to better clarify the discussion about the Gardner direct model. By calling $T_E$ the temperature below which the entropy of the paramagnetic phase becomes negative, it holds that in the limit of $p\to\infty$ a spin glass transition occurs exactly at $T_E$. As mentioned also in the following point this is consistent with the fact that in this limit the model converges to a Random Energy Model with temperature rescaled by $\sqrt{2\alpha}$. At finite $p$, we can only say that the model cannot be in the paramagnetic phase below $T_E$, therefore a spin glass transition should occur at a higher temperature. Since the RS spin-glass solution of Eqs. (2) exists only below $T_E$ (Fig. 1, violet region), the spin glass transition must be towards a RSB spin glass phase. We discuss this point at lines 131 - 139 in the new version of the manuscript.

2) We thank the referee for raising this point that we completely omitted in the overview of the Gardner direct model. We agree that it is better to add this discussion also in the paper. For $p>2$ and outside the signal retrieval phases the free energy is the same as for the $p$-spin glass model with Gaussian interactions (Gross and Mezard 1984, Gardner 1985), where the temperature is rescaled by a factor $\sqrt{2\alpha}$ and so spin-glass and paramagnetic solutions are the same as those found for this model. For finite $p$ (Gardner 1985), a 1RSB solution (with $m = k = 0$) exists and is globally stable throughout a whole phase below $T_s(p)\geq T_E$ but becomes unstable at a lower transition temperature $T_G(p)$. Below $T_G(p)$, one typically would expect multiple steps of RSB. In the limit of $p\to\infty$, it holds that $T_s(p)\to T_E$, $T_G(p)\to 0$ and the model becomes 1RSB, consistently with the fact that it is converging to a Random Energy Model (with rescaled temperature). We have clarified the final part of Section 2 accordingly. We discuss this point at lines 140 - 151 in the new version of the manuscript.

3) We cannot have $q^\star \neq 0$ and $m \neq 0$ at the same time as long as $T^\star>T_{\mathrm{crit}}$ because in this regime the examples are poorly correlated with the signal and solutions where the student pattern shares a macroscopic overlap with both of them do not exist. Conversely, when $T^\star<T_{\mathrm{crit}}$, examples are aligned with the signal and this type of (accurate) example retrieval region does appear. We make it more explicit at lines 226 - 228 in the new version of the manuscript.

4) We completely agree with the referee's comment. We didn't discuss the possible existence of a dynamical 1RSB transition in the Gardner direct model but one must expect it to exist since, as outlined before, outside the retrieval phases that model is equivalent to a $p$-spin glass model with Gaussian interactions. In this type of model a random first order transition (RFOT) phenomenology is observed (Monasson 1995, Montanari and Ricci Tersenghi 2003, Crisanti et al. 2005, Franz et al. 2017): there is in fact a region of temperatures, $T_s(p)\leq T\leq T_d(p)$ where the dynamics get trapped in an exponential number of metastable clusters, with an emerging RSB structure that doesn't affect the free energy. Below $T_s(p)$ the number of clusters is no longer exponential and the system undergoes a thermodynamic 1RSB phase transition This dynamical region coincides to the region in the inverse model where the signal retrieval state is locally stable. We added a discussion about this point in the new version of the manuscript at lines 151 - 160 and 322 - 324.

Requested changes: All of the requested changes were made. Reference 26 is now referred to as K & H's work instead of Krotov's work. See Fig. 2, lines 254-255 and the equation above them for the other changes.

---

## Round 1 · Referee Report · Anonymous (Referee 1) · 2024-5-3

Strengths

The manuscript "Dense Hopfield Networks in the Teacher-Student Setting" contains interesting and novel results, it is well-written, it is very thorough in the theoretical analysis, and validates the analytical results with monte-carlo simulations which largely agree with the theory. There are no major flaws in the paper that I can spot and overall it is a pleasant (although lengthy) read. The model reveals a rich phenomenology.

Weaknesses

  1. The central limitation of the study is in the teacher architecture, which is a very simple generalized hopfield model with only one memory. I understand that even considering a few memories would be technically challenging while having an extensive number of memories in the teacher would be well beyond current techniques. Could the authors give their opinion on this point? Do they have a hint on the scalings involved for M when one wants to recover K signals in the teacher

  2. The paper is somewhat lengthy, but not unnecessarily lengthy, it is really crammed with information that takes some time to parse. I'm not suggesting the paper should be shortened though.

Report

Intro

The paper analyzes the case of p-wise Hopfield model, from the point of view of spin glass systems. Theoretical results are obtained through the replica method in the RS ansatz. The setting is a teacher-student one, where the teacher is a similar architecture (but possibly mismatched in p and in temperature) with only a single memory. Given a certain number of samples produced by the teacher, bayes theorem (possibly under wrong assumptions by the statistician) is used to derive the posterior on the teacher's memory and this defines the student. A similar study has been done for the case p=p=2 and here generalized to arbitrary p,p. The authors thoroughly discuss the different resulting phases (paramagnetic, local retrieval, global retrieval, example retrieval), first considering a matched setting (Nishimori line) and then unmatched temperature and interaction order. Finally, in the p!=p* case where there is a coexistence of example and signal retrieval phases, the authors consider the effect of perturbations of the signal configuration in the direction of one of the examples, mimicking the adversarial attack framework of deep learning.

Given the quality of the work, I strongly recommend it for publication. I have no major perplexity about the paper, what follows are some minor comments.

  1. At the beginning of section 3 I think alpha = M p! / N^{p-1} as in previous section, and this is also true in eqs. (3) and (4). This should be made explicit, because at the end of Section 3 different scalings are considered. It would help readability to stress (again) that 3 and 4 apply for p=p* just before eq. 3.

  2. Should the list of phases in Section 3 also include the accurate example retrieval phase, where both m != 0 and q* != 0?

  3. Fig.2 could specify in the caption that this is the matched setting T=T and p=p. It would also help to point the reader that the vertical line can be desumed from the alpha=0 axis in Fig. 1.

  4. In caption Fig. 4 define epsilon to help readability

  5. I think I didn't fully understand the claim "the direct model is in the paramagnetic phase if and only if the inverse model is in the paramagnetic phase.", since it seems to me that it would be imply the orange region in Fig. 1 left would be the same as Fig 2 left, except for RSB effects and for the eR phase.

  6. In Section 4.5, the authors consider what is in some sense a weakly adversary setting. That is, they perturb in the direction of a randomly chosen example. This is a very reasonable direction of instability, but as they show with MC simulation a stronger adversary could be created by optimizing over the choice of the example, and possibly stronger ones could be created by optimizing over arbitrary directions. It would be interesting and I think very easy to complement the analysis showing the tolerance with respect to perturbation toward the paramagnetic state (i.e. uniform configuration), provided the P solution is locally stable in this region. This would set a "dummy" baseline over which "adversarial" attacks should improve.

Recommendation

Publish (surpasses expectations and criteria for this Journal; among top 10%)

  • validity: high
  • significance: high
  • originality: high
  • clarity: high
  • formatting: excellent
  • grammar: excellent

Author:  Robin Thériault  on 2024-05-21  [id 4502]

(in reply to Report 1 on 2024-05-03)
Category:
answer to question

We thank the referee for carefully reading the manuscript and for the insightful comments. As the referee proposed in the \textit{weaknesses} section, we added a few lines about learning more than one pattern (lines 550 - 553). We completely agree that this is a relevant addition as it better highlights how our results can be generalized. We also divided the text into smaller sections to hopefully improve readability. Concerning the list of remarks, a point-by-point answer follows:

Concerning points 1, 3 and 4, minor changes were made following the referee's suggestions. See lines 188 - 189, 204, 247 for point 1, lines 193 - 194, 200, 228 and Fig. 2 for point 3 and Fig. 4 for point 4.

2) We thank the referee for bringing to our attention that our phrasing was not very clear. The accurate example retrieval phase is not on the list of phases in section 3 because it enumerates the phases that we see when $T^* > T_{\mathrm{crit}}$. We made it clearer in the new version (lines 226 - 228).

5) One again, we thank the referee for bringing to our attention that our phrasing was not very clear. We meant to say that, when $T > T_{\mathrm{crit}}$, the inverse model is in the paramagnetic iff the direct model is in the paramagnetic phase. In the new version, we replaced the phrasing "outside of the eR phase" by "when $T > T_{\mathrm{crit}}$" for more clarity (lines 269, 277, 295, 296).

6) We think it is a very good idea to compare adversarial attacks to random perturbations to establish a baseline. As per the referee's suggestion, we compare adversarial attacks to random perturbations at lines 494 - 496 in the new version of the manuscript. To summarize, we find that random perturbations are much less efficient at fooling the model than adversarial attacks.

---

## Round 2 · Referee Report · Anonymous (Referee 1) · 2024-5-27

Report

I'm satisfied with the revised paper and recommend it for publication.

Recommendation

Publish (easily meets expectations and criteria for this Journal; among top 50%)

---

## Round 2 · Referee Report · Anonymous (Referee 2) · 2024-6-25

Report

I'm satisfied with the revised paper and recommend it for publication.

Recommendation

Publish (easily meets expectations and criteria for this Journal; among top 50%)

---

## Round 2 · List of Changes

Sections: Divided section 3 into 3.1 and 3.2. Split 4.1 into 4.1 and 4.2. Divided ex-section 4.4 (now 4.5 because of renumbering) into 4.5.1 and 4.5.2.

Line 51-53: Clarified a sentence in the introduction.

Line 83-85: Added a few lines describing additions to Section 2.

Line 136 - 160: Added a discussion about RSB, dynamical 1RSB and the spin-glass phase in the direct model.

Lines 188 - 189, 204, 247: Clarified the scaling of $\alpha$ at a few places.

Lines 193 - 194, 200, 228: Added references pointing to Fig. 1 to clarify the meaning of $T_{\mathrm{crit}}$.

Line 201: Added a reminder that $p^* = p$.

Lines 226 - 228: Explained why we cannot have $q^* \neq 0 $ and $m \neq 0$ at the same time.

Lines 269, 277, 295, 296: Clarified that we need $T > T_{\mathrm{crit}}$ for the argument of section 4.2 to hold.

Lines 321 - 324: Added a brief discussion about RSB and dynamical 1RSB in the inverse model (and how they relate to the direct model).

Lines 494 - 496: Added a comparison between adversarial attacks and random perturbations.

Lines 550 - 553: Added a few lines about learning more than one pattern.

Fig. 2: Specified that the plot is on the Nishomori line and added a reference pointing to Fig. 1 in order to explain how to find the $eR$ transition temperature from the $\alpha = 0$ axis. Added phase transition lines of nRSB where $n > 1$, 1RSB and d1RSB in the direct model. Modified the caption correspondingly.

Fig. 4: Defined $\varepsilon$.

Citations: Citation 26 is now referred to as K \& H's work instead of Krotov's work. Added seven references to support the new discussion at lines 136-160.

---

## Editorial Decision

published